# Extracellular Pgk1 enhances neurite outgrowth of motoneurons through Nogo66/NgR-independent targeting of NogoA

Cheng Yung Lin[1], Chia Lun Wu[2], Kok Zhi Lee[2], You Jei Chen[2], Po Hsiang Zhang[1], Chia Yu Chang[3,4], Horng Jyh Harn[3,5], Shinn Zong Lin[3,6]*, Huai Jen Tsai[1]*

[1]Institute of Biomedical Sciences, Mackay Medical College, New Taipei City, Taiwan; [2]Institute of Molecular and Cellular Biology, National Taiwan University, Taipei, Taiwan; [3]Bioinnovation Center, Buddhist Tzu Chi Medical Foundation, Hualien City, Taiwan; [4]Department of Medical Research and Neuroscience Center, Buddhist Tzu Chi General Hospital, Hualien City, Taiwan; [5]Department of Pathology, Buddhist Tzu Chi General Hospital and Tzu Chi University, Hualien City, Taiwan; [6]Department of Neurosurgery, Buddhist Tzu Chi General Hospital, Hualien City, Taiwan

*For correspondence:
shinnzong@yahoo.com.twn (SZL);
hjtsai@ntu.edu.tw;
hjtsai@mmc.edu.tw (HJT)

**Competing interests:** The authors declare that no competing interests exist.

**Abstract** NogoA inhibits neurite outgrowth of motoneurons (NOM) through interaction with its receptors, Nogo66/NgR. Inhibition of Nogo receptors rescues NOM, but not to the extent exhibited by *NogoA*-knockout mice, suggesting the presence of other pathways. We found that NogoA-overexpressing muscle cells reduced phosphoglycerate kinase 1 (Pgk1) secretion, resulting in inhibiting NOM. Apart from its glycolytic role and independent of the Nogo66 pathway, extracellular Pgk1 stimulated NOM by triggering a reduction of p-Cofilin-S3, a growth cone collapse marker, through decreasing a novel Rac1-GTP/p-Pak1-T423/p-P38-T180/p-MK2-T334/p-Limk1-S323/p-Cofilin-S3 molecular pathway. Not only did supplementary Pgk1 enhance NOM in defective cells, but injection of Pgk1 rescued denervation in muscle-specific NogoA-overexpression of zebrafish and an Amyotrophic Lateral Sclerosis mouse model, SOD1 G93A. Thus, Pgk1 secreted from muscle is detrimental to motoneuron neurite outgrowth and maintenance.
DOI: https://doi.org/10.7554/eLife.49175.001

## Introduction

The role of NogoA in limiting axonal fiber growth and regeneration following neuronal injury is well known (*Delekate et al., 2011*). NogoA inhibits neurite outgrowth of motoneurons (NOM) via Nogo66 and NiGΔ20 domains (*GrandPré et al., 2000*). Pharmaceutical and genetic strategies to ablate known receptors interacting with Nogo66 domain (*Fournier et al., 2001*; *Atwal et al., 2008*; *Nakamura et al., 2011*) have not successfully reversed inhibited NOM to the extent exhibited by *NogoA*-knockout mice (*Simonen et al., 2003*), suggesting the presence of other pathways.

In Amyotrophic Lateral Sclerosis (ALS) patients, overexpression of NogoA in muscle was positively correlated with the loss of motor endplates, and as the presence of NogoA in muscle cells increases, denervation of endplates also increases (*Bruneteau et al., 2015*). It has also been reported that patients whose NogoA was overexpressed in muscle cells and who were diagnosed with lower motor neuron syndrome (LMNS) also had a higher likelihood of progressing to ALS symptoms (*Pradat et al., 2007*). In a mouse, NogoA could be detected in the muscles of embryos at E15 and P1 stages, but not in the muscles in adults (*Wang et al., 2002*). Interestingly, a significant increase of NogoA was found in the muscle fibers of SOD1 G86R mice and ALS patients (*Dupuis et al.,*

*2002*; *Jokic et al., 2005*). *NogoA*-knockout in SOD1 G86R mice (*G86R/NogoA$^{-/-}$*) showed longer survival time and increased survival rate (*Jokic et al., 2006*). Moreover, when NogoA was overexpressed in the muscle of mice, morphological changes of neuromuscular junction (NMJ) were also observed (*Jokic et al., 2006*). More recently, in a zebrafish model, overexpression of Rtn4al (human NogoA homolog) in muscle exhibited an ALS-like phenotype (*Lin et al., 2019*). This line of evidence suggests that overexpression of NogoA/Rtn4a in muscle cells may contribute to the etiology of ALS-like disease at an early stage, even though NogoA could not be detected in the blood (*Harel et al., 2009*). However, the molecular mechanism that would explain the association between increased NogoA in muscle cells and the occurrence of neurodegeneration remains largely unclear. Therefore, it is a plausible hypothesis that the inhibitory effect of NogoA-overexpressed muscle cells on NOM may be mediated through some secretory myokine released by muscle cells.

Therefore, to gain a better understanding of NogoA overexpression in muscle relative to the effects of myokine secretion, we first collected conditional medium (CM) from cultured Sol8 myogenic cells in which NogoA is overexpressed (Sol8-NogoA CM). Next, we incubated mouse motoneuron hybrid NSC34 cells in Sol8-NogoA CM and found appreciable inhibition of NOM. Then, we found that the amount of secreted phosphoglycerate kinase 1 (Pgk1) was dramatically reduced in Sol8-NogoA CM compared to control Sol8 CM. Upon addition of Pgk1, NOM derived from NSC34 cells cultured in Sol8-NogoA CM was rescued. Furthermore, we found that extracellular Pgk1 (ePgk1) reduces the phosphorylation of Pak1/P38/MK2/Limk1/Cofilin axis, which, in turn, enhances the NOM in a manner functionally independent from its intracellular, canonical role as a supplier of energy. Therefore, we conclude that reduced secretion of Pgk1 from NogoA-overexpressed muscle cells inhibits NOM promises a paradigm shift that will inspire new thinking about therapeutic targets against failure NOM and denervated NMJ.

## Results

### Supplementary addition of Pgk1 improves neurite outgrowth of NSC34 cells cultured in CM obtained from cultured myoblast cells expressing Sol8-NogoA

The inhibition of NOM by muscle-specific NogoA might result from inducing a canonical Nogo66/NgR or Nogo66/PirB signaling pathway or from influencing the secretion of myokines. To define this molecular mechanism, we employed mouse Sol8 myoblasts and established a Doxycycline (Dox)-inducible cell line stably harboring the Sol8-vector only (Sol8-vector) and the Sol8 vector with NogoA insert (Sol8-NogoA) (*Figure 1—figure supplement 1*). After induction, the CM obtained from cultured myoblast cells harboring Sol8-vector was directly used to culture motoneuron cell line NSC34. After culturing, motoneurons exhibited extended neurites, while motoneurons incubated with Sol8-NogoA CM failed to exhibit neurite outgrowth (*Figure 1—figure supplement 1*). Importantly, NogoA was not detected in the CM cultured by Sol8-NogoA cells after Dox induction for 48 hr (*Figure 1—figure supplement 1C*). This line of evidence suggested that the component(s) contained in CM from cultured Sol8-NogoA play(s) a role in NOM inhibition, but not through NogoA contained in medium. Moreover, unlike neurons treated with Sol8-vector, Western blot analysis demonstrated that the signal intensity of phosphorylated Cofilin at S3 (p-Cofilin-S3), which is a hallmark of growth cone collapse in neuronal cells, as well as a marker of reduced axonal actin dynamics in ALS patients with depleted and mutated C9ORF72 (*Heredia et al., 2006*; *Sivadasan et al., 2016*), was upregulated in neurons treated with Sol8-NogoA CM (*Figure 1—figure supplement 1*). Collectively, this line of evidence suggested that the component(s) contained in CM from cultured Sol8-NogoA play(s) a role in NOM inhibition.

We further employed 2D electrophoresis, followed by LC MS/MS, to analyze the total proteins content in CM. Among these examined proteins, we found that the level of Pgk1 protein contained in Sol8-NogoA CM was significantly reduced compared to that in Sol8-vector CM (see *Figure 1—figure supplement 2*). Furthermore, the degree of p-Cofilin expression in NSC34 cells cultured in CM from Pgk1-overexpressed Sol8-NogoA cells was significantly reduced compared with that in NSC34 cells cultured in CM from Sol8-NogoA cells. Therefore, Pgk1 was chosen for further study to confirm its potential role in NOM. Additionally, we used Western blot analysis to detect the protein level of Pgk1 in the sera of transgenic SOD1 G93A mice, in which NogoA was overexpressed in muscle

(*Figure 1—figure supplement 3*), the results of which were consistent with those reported by *Bros-Facer et al. (2014)*. We found that the amount of Pgk1 in the sera of SOD1 G93A mice was significantly reduced compared with that in the sera of WT mice (*Figure 1—figure supplement 3*). Collectively, based on the above results provided *in vitro* and *in vivo* evidence, we chose Pgk1 for further study to confirm its potential role in NOM.

We added Pgk1 directly into Sol8-NogoA CM and found that p-Cofilin-S3 in NSC34 cultured with Sol8-NogoA CM plus Pgk1 was lower than that of cells cultured with Sol8-NogoA CM (*Figure 1A*). Importantly, supplementary Pgk1 could reduce p-Cofilin-S3 to then restore NOM of NSC34 cells, even though they had been cultured in Sol8-NogoA CM, indicating that stalled NOM of NSC34 cells cultured in Sol8-NogoA CM could be rescued upon the addition of Pgk1.

To further confirm whether Pgk1 secreted by muscle cells could enhance NOM, we added Pgk1-specific antibody to Sol8-vector CM and then cultured NSC34 cells. In response, NOM was totally inhibited and p-Cofilin-S3 was higher compared to those observed in the control groups, such as Sol8-vector CM and Sol8-vector CM plus rabbit IgG (*Figure 1B*). Thus, we suggest that Pgk1 is secreted by myoblasts and that it exists in muscle cell CM to play an important role in enhancing NOM.

Next, to determine if Pgk1 alone is sufficient to induce NOM and neuronal differentiation, we switched to low trophic factor differentiation media (DM). Under the same condition, Pgk1 was able to extend NOM and downregulate p-Cofilin-S3 (*Figure 1C*). We also found that the effect of adding Pgk1 on the downregulation of p-Cofilin-S3 was dose-dependent from 0, 11, 33, 66 and 99 ng/ml (*Figure 1—figure supplement 4*). Moreover, when NSC34 cells were treated with Pgk1 at 33 ng/ml for 0, 8, 16, 24 and 48 hr, we found that the effect of Pgk1 on the downregulation of p-Cofilin-S3 was dose-dependent from 8 through 24 hr (*Figure 1—figure supplement 4*). However, we noticed that p-Cofilin-S3 was unexpectedly increased if cells were treated with Pgk1 for 48 hr. The doses of Pgk1 we examined did not cause any negative effect on cell growth and survival. However, based on the above results and to ensure that we did not seriously affect cell physiology, we treated NSC34 cells with the minimal concentration (33 ng/ml of Pgk1 for 24 hr) throughout the entire study. Interestingly, the addition Pgk1 at that concentration not only rescued the number of neurite-bearing cells, but also enhanced neurite length of NSC34 cells cultured in Sol8-NogoA CM and DM (*Figure 1D–E*).

## Supplementary addition of Pgk1 can enhance the maturation of NSC34 cells cultured in Sol8-NogoA CM

To further learn if these restored neurites in NSC34 cells could differentiate into mature neurons, we employed antibody combined with fluorescence staining to specifically detect Syn1, which is a marker labeling neuronal synaptic vesicles in cell terminals (*Fletcher et al., 1991*), and maturation markers, such as MAP2, GAP43 and ChAT (*Maier et al., 2013*). Based on the distribution by immunofluorescence staining against Syn1 within NSC34 cells cultured in CM, the percentages of NSC34 cells displaying the Syn1 signal in the growth cones terminal were 43 ± 4.78, 13 ± 1.63 and 45 ± 4.55% in Sol8-vector CM, Sol8-NogoA CM and Sol8-NogoA CM plus Pgk1, respectively (*Figure 2A–D*). The number of neurons with Syn1 signal at the end of growth cones could be restored to that cultured in Sol8-vector CM after Pgk1 was added in Sol8-NogoA CM.

We next examined the expression levels of functional proteins, such as MAP2, GAP43, Syn1 and ChAT, in NSC34 cells. The expressions of MAP2, GAP43, Syn1 and ChAT were all increased when Pgk1 was added to the Sol8-NogoA CM/NSC34 culture (*Figure 2E*), suggesting that Pgk1 addition could rescue weak cell differentiation caused by culturing in Sol8-NogoA CM. Taken together, we conclude that exogenous addition of Pgk1 enables NSC34 cells cultured in Sol8-NogoA CM to improve differentiation, maturation, and neurite growth.

## Reduced protein level of p-Cofilin-S3, as mediated by ePgk1, is independent of glycolysis

Since Pgk1 is a key enzyme for glycolysis, it is necessary to determine whether reduced p-Cofilin-S3 and resultant enhancement of NOM by addition Pgk1 could be attributed to glycolytic effect. Neither the absence nor presence of ePgk1 resulted in any significant difference in glycolytic function of neurons (*Figure 3A*). Even in the low glycolytic condition induced by glycolysis inhibitor 3-(3-

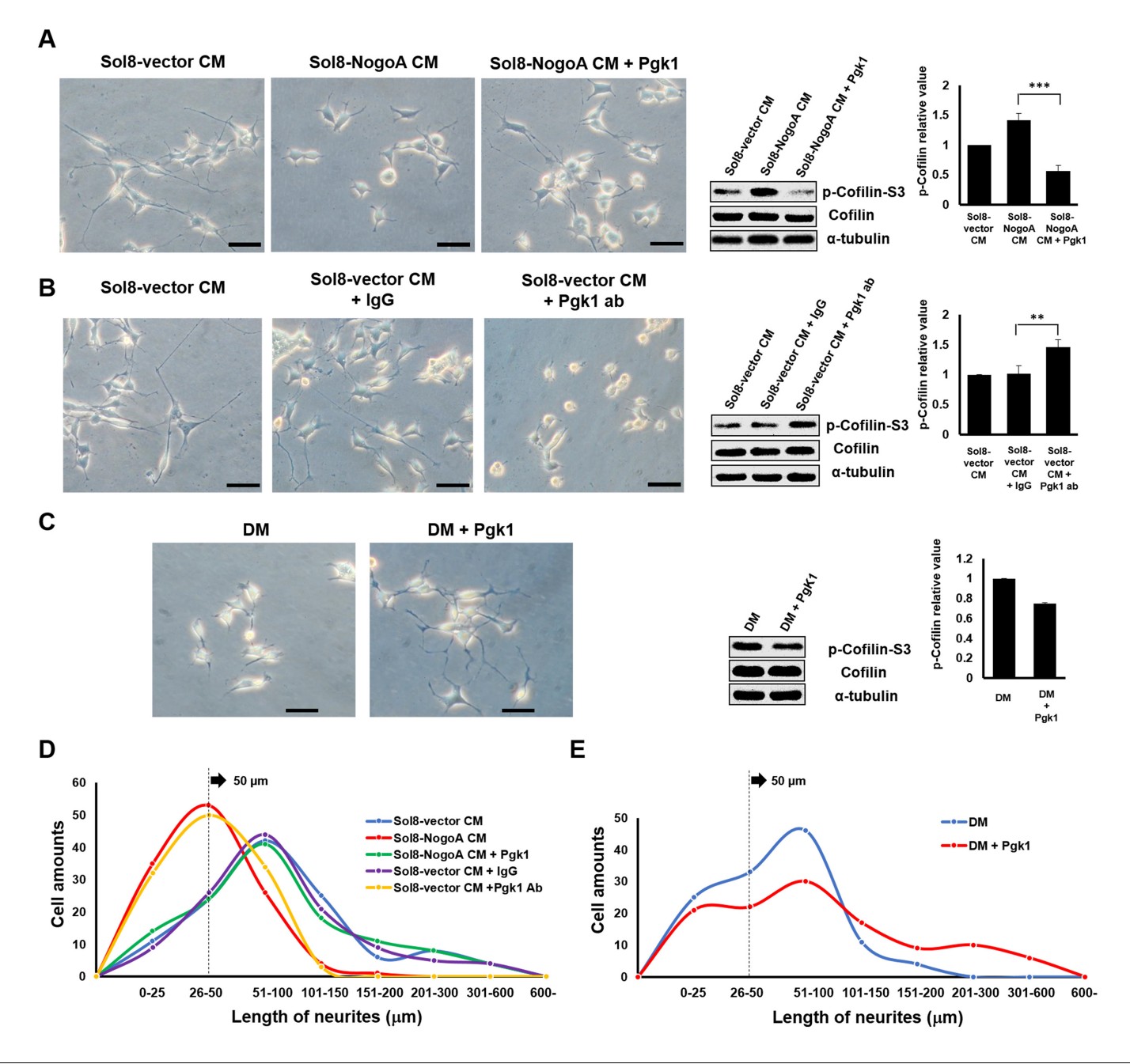

**Figure 1.** Supplementary addition of Pgk1 in culture medium promotes neurite outgrowth developed from NSC34 cells. (**A**) Neurite outgrowth of motoneurons developed from NSC 34 motoneuron cells cultured in either Sol8-vector CM or Sol8-NogoA CM with and without Pgk1 addition. (**B**) Neurite outgrowth and p-Cofilin-S3 expression in NSC34 cells cultured in Sol8-vector CM without Pgk1. Rabbit IgG (control) and Pgk1 antibody were separately added into Sol8-vector CM. (**C**) Neurite outgrowth of motoneurons developed from NSC34 cells cultured in differentiation media (DM) with or without Pgk1 addition. (**A-C**) Right panels: Western blot analysis of total Cofilin and p-Cofilin-S3 contained in NSC34 cells. Statistical analysis used Student's $t$-test (***, $p < 0.001$; **, $p < 0.01$). (**D–E**) The patterns of neurite length distribution. Cell number with various lengths of neurites of NSC 34 cells cultured by (**D**) CM obtained from Sol8-vector, Sol8-NogoA with or without Pgk1 addition, Sol8-vector with IgG and with Pgk1 antibody addition, as indicated, and by (**E**) DM with or without containing Pgk1 was determined (120 cells per experimental condition).

DOI: https://doi.org/10.7554/eLife.49175.002

The following figure supplements are available for figure 1:

**Figure supplement 1.** Conditioned media used to culture myoblasts overexpressing NogoA inhibited the neurite outgrowth of motor neurons.

DOI: https://doi.org/10.7554/eLife.49175.003

*Figure 1 continued on next page*

*Figure 1 continued*

**Figure supplement 2.** Comparison of protein patterns of conditioned medium from culturing myoblasts with and without overexpressing NogoA.
DOI: https://doi.org/10.7554/eLife.49175.004
**Figure supplement 3.** Using Western blot analysis to detect the protein level of Pgk1 in serum of mouse.
DOI: https://doi.org/10.7554/eLife.49175.005
**Figure supplement 4.** The expression of p-Cofilin-S3 in NSC34 cells treated with different doses of Pgk1 and a time course induction.
DOI: https://doi.org/10.7554/eLife.49175.006

Pyridinyl)−1-(4-pyridinyl)−2-propen-1-one (3PO) (*Figure 3B*), in which p-Cofilin-S3 was insignificant compared to the untreated group (*Figure 3C*, lanes 1 vs. 3), ePgk1 still downregulated p-Cofilin-S3 (*Figure 3C*, lanes 3 vs. 4). Moreover, addition of Pgk1 mutants, Pgk1-T378P and Pgk1-D375N, both

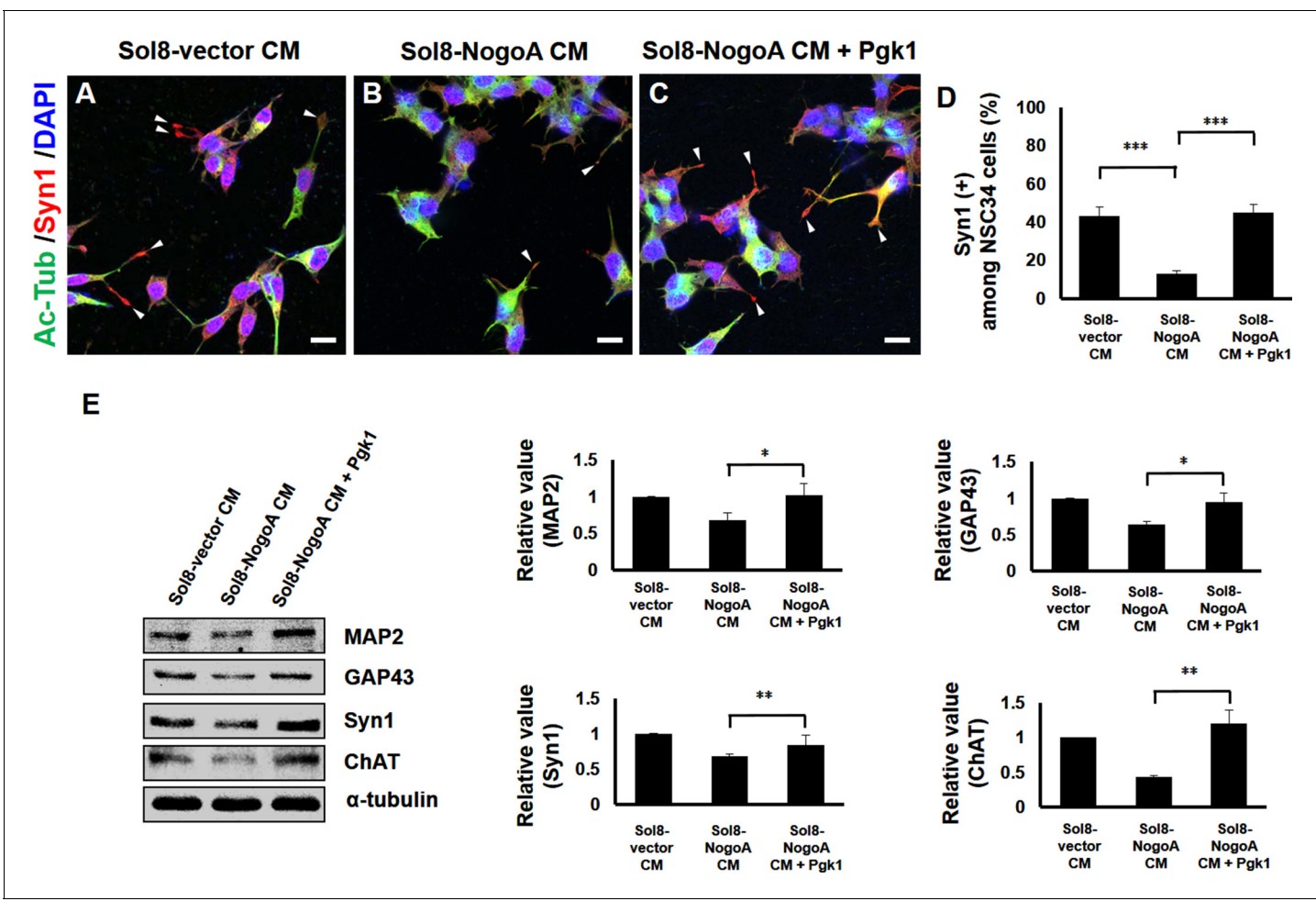

**Figure 2.** Supplementary addition of Pgk1 enables more NSC34 cells cultured in Sol8-NogoA CM to differentiate into mature motor neurons. Immunofluorescence staining of NSC34 cells cultured in (A) Sol8-vector CM, (B) Sol8-NogoA CM, and (C) Sol8-NogoA CM plus Pgk1. DAPI was labeled by blue fluorescent signal to mark nucleus, and acetyl-tubulin was labeled by green signal, while Syn1 was labeled by red signal. The growth cone of NSC34 cells was marked with white arrowheads. Scale bar, 20 µm. (D) In total, 110 to 120 cells were randomly selected from each culture group, and the number of Syn1-positive cells with growth cone was counted as a percentage. (E) Western blotting analysis was performed to quantify the expression levels of MAP2, GAP43, Syn1 and ChAT from three treatment groups. α-tubulin served as internal control. The relative value of each examined protein was used for comparison among the three groups when the value obtained from the Sol8-vector CM group was normalized as 1. Data were averaged from three independent experiments. Statistical analysis used Student's *t*-test (***, significant difference at p<0.001; **, p<0.01; *, p<0.05).
DOI: https://doi.org/10.7554/eLife.49175.007

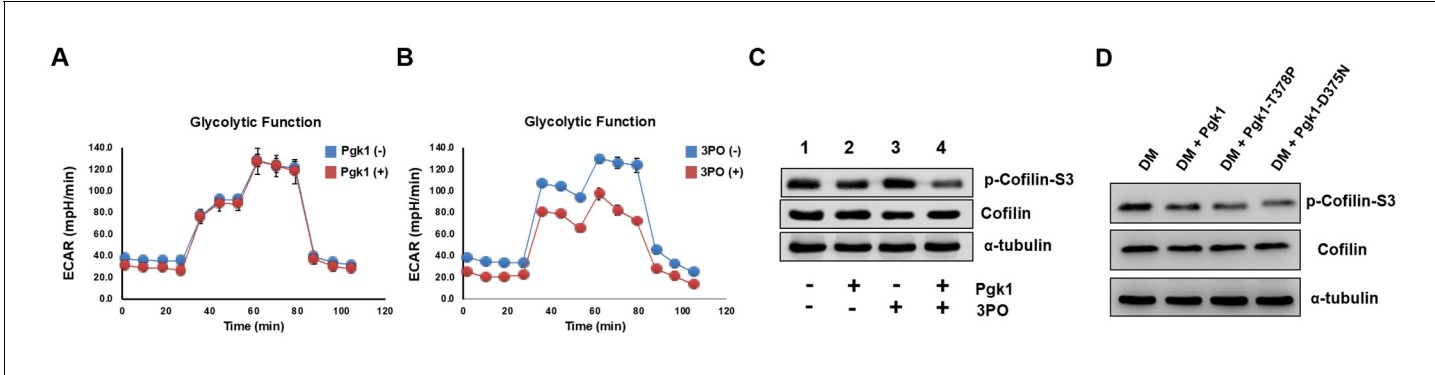

**Figure 3.** Enhancement of neurite outgrowth, as mediated by supplementary addition of Pgk1, is independent of metabolic glycolysis. (A) Analysis of glycolytic function in NSC34 cells cultured in DM with or without the addition of Pgk1 or in (B) glycolysis inhibitor 3PO. (C) Western blot analysis of total Cofilin, p-Cofilin-S3, and protein control marker (α-tubulin) of NSC34 cells cultured in conditions, as indicated. (D) Western blot analysis of total Cofilin, phosphorylated Cofilin (p-Cofilin-S3), and protein control marker (α-tubulin) of NSC34 cells cultured in DM containing Pgk1 and its catalytic mutants, Pgk1-T378P and Pgk1-D375N. All of the above data were averaged from three independent experiments.
DOI: https://doi.org/10.7554/eLife.49175.008

of which lost their catalytic activities (*Lay et al., 2002*; *Chiarelli et al., 2012*) in DM, could still reduce p-Cofilin-S3 (*Figure 3D*), suggesting ePgk1 promotes NOM independent of glycolysis.

## Specific knockout of Pgk1 in zebrafish muscle cells causes defective development of motoneurons

To confirm the role of Pgk1 secreted from muscle cells regulates motoneuron development *in vivo*, we employed the CRISPR/Cas9 system to knock out Pgk1 in zebrafish embryos. We used muscle-specific α-actin promoter to drive overexpression of Cas9 in muscle cells in order to exclusively knock out the *pgk1* gene. Since turbo-red fluorescent protein (tRFP) was engineered to fuse with Cas9 and P2A peptide, it served as a reporter to reflect the overexpression of Cas9 (*Figure 4A*). Compared to control *Tg(mnx:GFP)* embryos at 30 hpf (*Figure 4C*), the tRFP signal was observed in the muscle of pZα-Cas9-injected embryos, indicating that Cas9 was overexpressed in certain muscle cells (*Figure 4D*). Nevertheless, motoneurons were normally developed, which suggests that overexpression of Cas9 in muscle cells had no effect on development. However, when *Tg(mnx:GFP)* embryos were coinjected with pZα-Cas9 and *pgk1* sgRNA, which inhibits the production of Pgk1 in muscle cells (*Figure 4—figure supplement 1*), defective motoneurons were observed (*Figure 4E*), suggesting that the reduction of Pgk1 in muscle cells is followed by impairment of NOM. In a parallel experiment, by conditional knockout of *phosphoglycerate mutase 2* (*pgam2*) in muscle cells (*Figure 4—figure supplement 1*), a downstream enzyme of Pgk1 in the glycolytic pathway, we again confirmed that the glycolysis pathway in muscle did not affect NOM (*Figure 4F*). Collectively, these results suggested that defects in NOM result from a decrease in Pgk1 secretion in muscle cells. To prove the consistency of this result *in vivo*, we constructed a plasmid containing Pgk1 fused with P2A peptide and tRFP (*Figure 4B*). After injecting this construct into zebrafish embryos, RFP exhibited specifically in muscle cells. When these RFP-expressing cells were sorted out and examined by Western blot analysis, the results demonstrated that Pgk1 was overexpressed (*Figure 4—figure supplement 1*). Meanwhile, motoneuron axons exhibited ectopic growth (*Figure 4G*). As suggested by quantification of the axonal extension phenotype shown in *Figure 4—figure supplement 2*, the increase of Pgk1 in muscle cells enhances NOM.

## ePgk1 reduces the p-Cofilin-S3 expressed in NSC34 cells by decreasing the phosphorylation of Limk1 at S323

We next sought to clarify the molecular pathway that allows addition of Pgk1 to promote neuronal differentiation and NOM of NSC34 cells. The binding of Nogo66 domain to Nogo receptor (NgR) induces an acute intracellular signaling response to inhibit NOM through (i) increasing p-Cofilin-S3 (*Ohashi et al., 2000*; *Manetti, 2012*) by phosphorylating Limk1 (p-Limk1-T508) in the Rho/ROCK

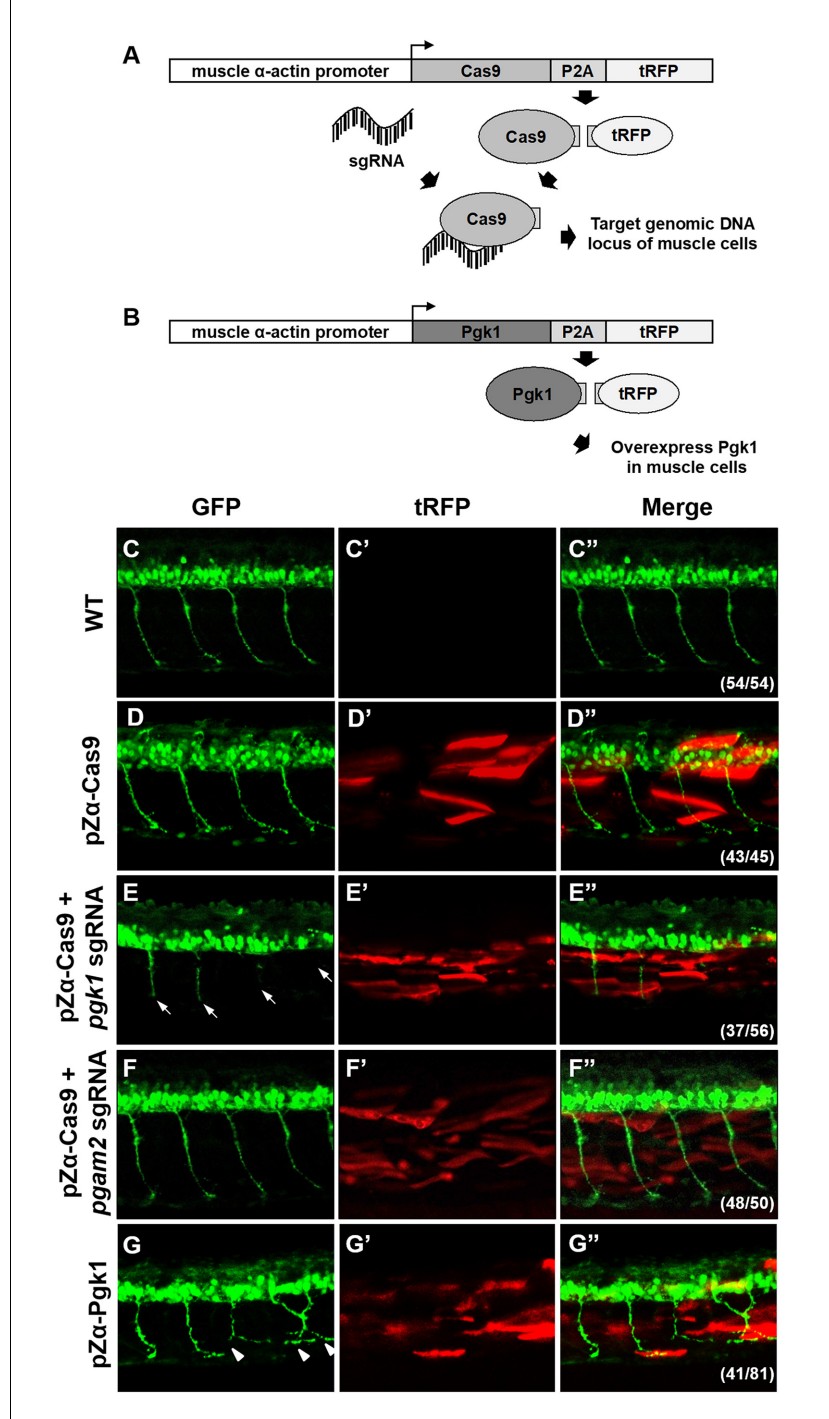

**Figure 4.** Muscle-specific overexpression of Pgk1 enhances NOM in zebrafish embryos. (A) Diagram of plasmid pZα-Cas9. The Cas9-P2A-tRFP cassette is driven by zebrafish muscle-specific α-actin promoter. The fusion protein Cas9-P2A-tRFP expressed in muscle cells is digested into Cas9, which is bound by transferred sgRNA, resulting in silencing the target gene in muscle cells. (B) Diagram of plasmid pZα-Pgk1. Pgk1 is specifically overexpressed in muscle cells. (C–G) Injection of different materials, as indicated, into embryos from transgenic line *Tg(mnx:GFP)* and observation of fluorescent signals expressed in embryos at 30-hpf. GFP-labeled motor neurons observed under confocal microscopy. (C'–G') Location of RFP-labeled muscle cells in which Cas9 and/or Pgk1 is overexpressed. (C''–G'') Two fluorescent signals were merged. Numbers shown in the lower right corner were the number of phenotypes out of total examined embryos. (C–C'') Untreated embryos served as the control group. (D–D'') Injection of pZα-Cas9. NOM was not affected. (E–E'') Injection pZα-Cas9 combined with *pgk1* sgRNA. The

*Figure 4 continued on next page*

*Figure 4 continued*

length of NOM became shorter (white arrows). (F–F") Injection of pZα-Cas9 combined with *pgam2* sgRNA (served as negative control). The NOM was not affected. (G–G") Injection of pZα-Pgk1. The NOM became increasingly ectopic toward the muscle cells in which Pgk1 was overexpressed (white arrowheads).

DOI: https://doi.org/10.7554/eLife.49175.009

The following figure supplements are available for figure 4:

**Figure supplement 1.** Western blot analysis to detect Pgk1 protein level in the muscle of zebrafish embryos.

DOI: https://doi.org/10.7554/eLife.49175.010

**Figure supplement 2.** Quantitative analysis demonstrating the effect of Pgk1 expressed in zebrafish muscle on the growth of axonal motor neurons.

DOI: https://doi.org/10.7554/eLife.49175.011

pathway, (ii) increasing phosphorylated EGFR (p-EGFR-Y1173) (*Koprivica et al., 2005*), or (iii) down-regulating AKT phosphorylation (p-AKT-S473) (*Wang et al., 2011*). Here, we demonstrated there was no significant changes of p-ROCK2-Y256, p-EGFR-Y1173 or p-AKT-S473 expressions among NSC34 cells cultured in Sol8-vector CM, Sol8-NogoA CM and Sol8-NogoA CM plus Pgk1 (*Figure 5—figure supplement 1*). Surprisingly, Sol8-NogoA CM plus Pgk1 did reduce p-Cofilin-S3 *via* p-Limk1-S323, but not p-Limk1-T508 (*Figure 5A*), suggesting that the ePgk1-mediated pathway is independent of the Nogo66/NgR/ROCK2-Y256/Limk1-T508 axis.

The overexpression of Limk1, Limk1-S323A and Limk1-T508V in NSC34 cells cultured in DM resulted in increasing the basal level of intracellular p-Cofilin-S3 (*Figure 5B*). When we treated cells overexpressing Limk1, Limk1-S323A, and Limk1-T508V with Pgk1, the cells overexpressing Limk1 still responded to Pgk1, as evidenced by decreased p-Cofilin-S3 (*Figure 5C*). Overexpression of Limk1-S323A, but not Limk1-T508V, abolished Pgk1-mediated p-Cofilin-S3 downregulation (*Figure 5C*), again confirming that Pgk1 decreased p-Cofilin-S3 through Limk1-S323, but not Limk1-T508.

## The signaling pathway underlying the involvement of ePgk1-mediated reduction of p-Cofilin-S3

We further identified upstream kinases, including P38/MK2 and Rac1/Pak1 (*Figure 5D*), along with their phosphorylated sites, using Sol8-vector CM and Sol8-NogoA CM in the absence and presence of Pgk1. Pak1 overexpression confirmed this series of signaling transduction pathway, consistent with our results using Sol8-vector CM and Sol8-NogoA CM (*Figure 5E*). On the other hand, when NSC34 cells were cultured in DM plus a Pak1 inhibitor, FRAX597, we found that the intracellular p-Pak1 was reduced and that the levels of p-P38-T180, p-MK2-T334, p-Limk-S323 and p-Cofilin-S3 were also reduced (*Figure 5—figure supplement 2*), suggesting that P38, MK2, Limk, and Cofilin are all downstream effectors of Pak1. In sum, ePgk1 triggers a reduction in p-Cofilin-S3, in turn promoting NOM through decreasing a novel p-Pak1-T423/p-P38-T180/p-MK2-T334/p-Limk1-S323 axis via reducing Rac1-GTP activity in neuronal cells.

To confirm whether Pgk1 involved regulating pathway is independent of activation of NgR receptor by NogoA. First, we determined whether the p-Cofilin expression is different between GST-Nogo66 addition and GST-Nogo66 plus Pgk1 addition into NSC34 cultured in DM. Compared to the control group in which GST was added in DM, NSC34 cells cultured in GST-Nogo66-added DM exhibited the increase of p-ROCK2-Y256 and p-EGFR-Y1173 expression and the decrease of p-Akt-S473 (*Figure 5—figure supplement 3A*), suggesting GST-Nogo66 addition did activate NgR response pathway. Interestingly, the increased p-ROCK2-Y256 and p-EGFR-Y1173 and the decreased p-Akt-S473 remained unchanged in NSC34 cultured in GST-Nogo66 with Pgk1 addition (*Figure 5—figure supplement 3A*), suggesting the addition of Pgk1 did not affect Nogo66/NgR pathway. However, compared to GST-Nogo66-added cells, the GST-Nogo66 plus Pgk1-treated NSC34 cells exhibited the reduction of p-Cpfilin-S3 expression through decreasing p-Limk-S323 but not through p-Limk-T508 (*Figure 5—figure supplement 3B*), suggesting the reduced expression of p-Cofilin is due to the presence of Pgk1 through pathway other than Nogo66/NgR interaction.

Second, we determined whether the p-Cofilin-S3 expression is still reduced in NSC34 by addition of Pgk1 into NSC34 cultured in DM when NgR receptor is blocked by NAP2, a NgR receptor antagonist peptide. The results showed that addition of NAP2 in culture led to the reduction of ROCK2 expression in NSC34 cells, which was consistent with the result reported by *Sun et al. (2016)*

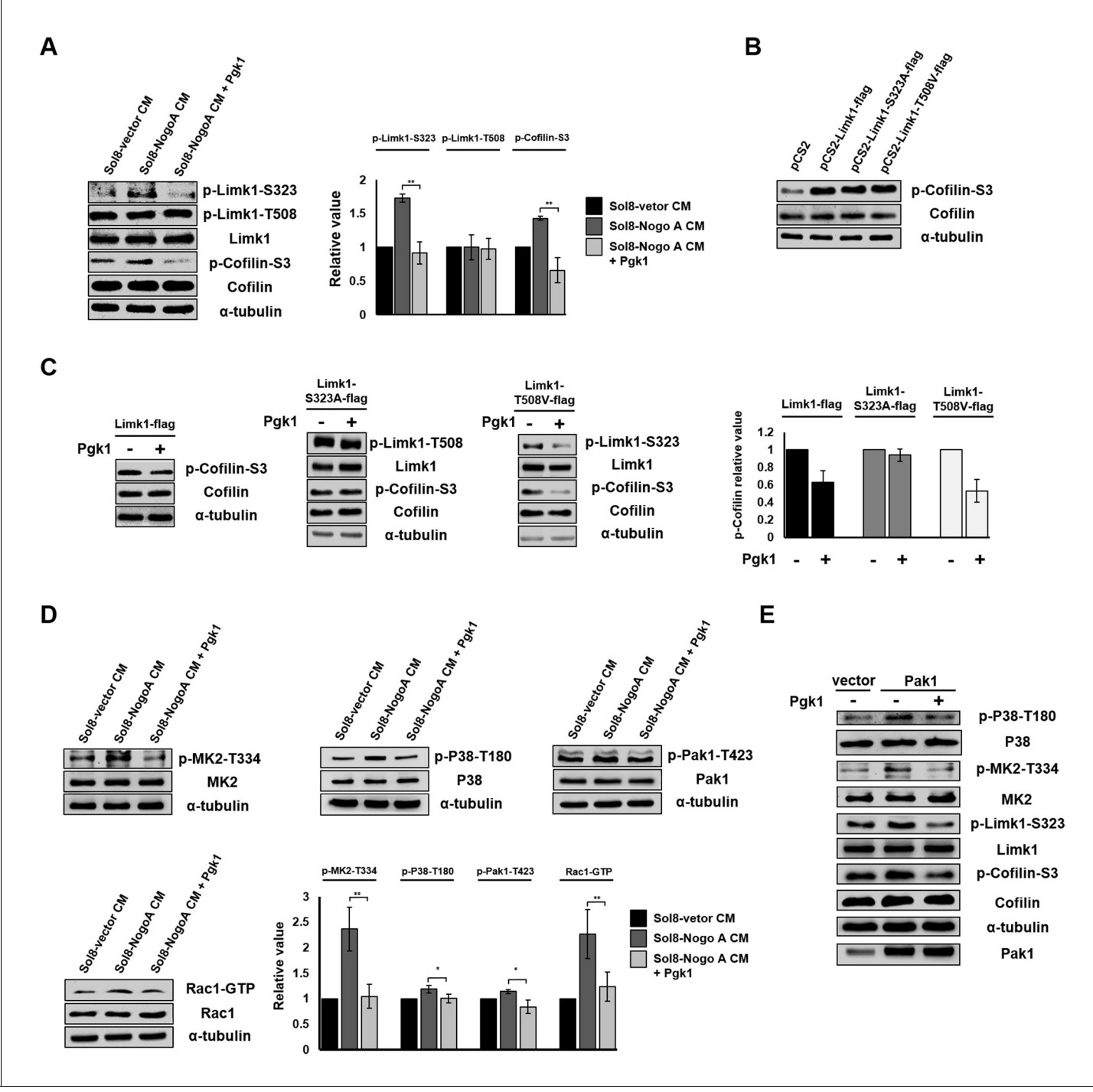

**Figure 5.** Pgk1 reduces the protein level of p-Cofilin-S3 through the decrease of phosphorylated Limk1 at S323 in NSC34 cells. (**A**) The expression levels of p-Limk1-S323, -T508, Limk1, p-Cofilin-S3 and total Cofilin in NSC34 cells treated with condition, as indicated. (**B**) Phosphorylated (p-Cofilin-S3) and total Cofilin contained in cells were examined by Western blot analysis. NCS34 cells cultured in DM were introduced separately with a plasmid of pCS2+, pCS2-Limk1-flag (expressing normal Limk1), pCS2-Limk1-S323A-flag (expressing S323A-mutated Limk1), and pCS2-Limk1-T508V-flag (expressing T508V-mutated Limk1). (**C**) Plasmids encoding overexpression of Limk1-, Limk1-S323A- and Limk1-T508V-flag fusion proteins were separately introduced into NSC34 cells cultured in DM with or without Pgk1. (**D**) Expression of P38/MK2 and Rac1/Pak1 markers of NSC34 cells cultured in condition, as indicated. (**E**) Detection of P38/MK2 and Limk-S323/Cofilin signaling pathway markers of NSC34 cells introduced with Pak1-overexpressing plasmid and cultured in DM with or without Pgk1. α-tubulin served as an internal control. Relative value represents the ratio of phosphorylated form value over total protein values. All data were averaged from three independent experiments with statistical analysis by Student's *t*-test (***, significant difference at p<0.001; **, p<0.01; *, p<0.05).

*Figure 5 continued on next page*

*Figure 5 continued*

DOI: https://doi.org/10.7554/eLife.49175.012

The following figure supplements are available for figure 5:

**Figure supplement 1.** The molecular pathway mediated by extracellular addition of Pgk1 to enhance neurite outgrowth is independent from the neuronal Nogo66/NgR Pathway.
DOI: https://doi.org/10.7554/eLife.49175.013

**Figure supplement 2.** Treatment of Pak1 inhibitor can reduce the phosphorylation levels of Pak1/P38/MK2/Limk1/Cofilin axis within NSC34 cells.
DOI: https://doi.org/10.7554/eLife.49175.014

**Figure supplement 3.** The reduction of p-Cofilin mediated by ePgk1 does not occur through Nogo66/NgR interaction in neuronal cells.
DOI: https://doi.org/10.7554/eLife.49175.015

indicating that NAP2 is able to block NgR receptor (*Figure 5—figure supplement 3C*). However, unlike the increased ROCK2 in NSC34 cultured in NAP2 addition, the ROCK2 expression remained unchanged in NSC34 cells cultured in NAP2 plus Pgk1 addition (*Figure 5—figure supplement 3C*). On contrast, the expression of p-Pak1-T423/p-P38-T180/p-MK2-T334/p-Limk1-S323/p-Cofiln-S3 axis was reduced (*Figure 5—figure supplement 3D*).

Taken together, we suggest that the reduction of p-Cofilin mediated by ePgk1 is not through the Nogo66/NgR interaction in neuronal cells since ePgk1 addition can be still functional to reduce p-Pak1/p-P38/p-MK2/p-Limk1-S323/p-Cofiln axis without the presence of NgR receptor of neuronal cells. In sum, ePgk1 triggers a reduction in p-Cofilin-S3, in turn promoting NOM through decreasing a novel p-Pak1-T423/p-P38-T180/p-MK2-T334/p-Limk1-S323 axis via reducing Rac1-GTP activity in neuronal cells. And, the molecular pathway ePgk1 involved in NOM is independent of the Nogo66/NgR interaction.

## Intramuscular injection of Pgk1 is able to rescue NMJ denervation caused by Rtn4al-overexpression transgenic zebrafish and SOD1 G93A transgenic mice

Muscle-specific NogoA-overexpression has been shown to disassemble NMJ in animal models, exhibiting muscle atrophy and reduced movement (*Jokic et al., 2006*; *Lin et al., 2019*). Noting the colocalization of the presynaptic markers synaptic vesicle glycoprotein 2A (SV2) and Neurofilament-H (NF-H), as well as the postsynaptic marker acetylcholine receptor labeled with α-Bungarotoxin (α-BTX), we asked if ePgk1 could re-establish NMJ integrity after its disassembly by NogoA overexpression. While we observed denervation at NMJ after Rtn4al/NogoA induction in adult zebrafish (*Figure 6—figure supplement 1*), as evidenced by reduced colocalization of presynaptic and postsynaptic markers, this defect saw much improvement by addition of Pgk1, but not GFP control, through intramuscular injection (*Figure 6A–G*), suggesting ePgk1 supports NMJ integrity. Although adult zebrafish muscle injected with Pgk1 did exhibit delayed NMJ denervation, it could still retain its normal biological activity for two weeks after a single shot (*Figure 6—figure supplement 2*).

We then used an ALS cell line and mouse model to determine any beneficial effects of Pgk1 supplementation. We first differentiated motor neurons derived from human induced pluripotent stem cells (iPSCs) harboring human SOD1 mutant (G85R) (*Figure 6—figure supplement 3*) and then added Pgk1. Pgk1 addition reduced p-Cofilin through decreasing p-Limk1-S323 (*Figure 6—figure supplement 4*), as noted above, indicating that ePgk1 induces the same signal transduction pathway as that determined from the mouse cell line and human motor neurons.

Next, to continue exploring this question *in vivo*, we performed intramuscular injection of Pgk1 into the gastrocnemius muscle of the right hind leg of 60-day-old transgenic SOD1 G93A mice, followed by another injection every 15 days until mice were 120 days old. We quantified the proportion of innervated NMJ in the gastrocnemius muscle of the right hind leg of mice. Compared to the control group, the proportion of innervated NMJ of WT mice injected with PBS (WT/PBS), which was normalized as 100%, the proportion of innervated NMJ of SOD1 G93A/PBS was $13 \pm 0.02\%$, indicating that the signal of motor neuron axon and axon terminal (NF-H/SV2) was significantly reduced. However, the proportion of innervated NMJ of SOD1 G93A mice injected with Pgk1 (SOD1 G93A/Pgk1) was $57 \pm 0.08\%$, suggesting that supplementary addition of Pgk1 could rescue NMJ denervation (*Figure 6H–N*).

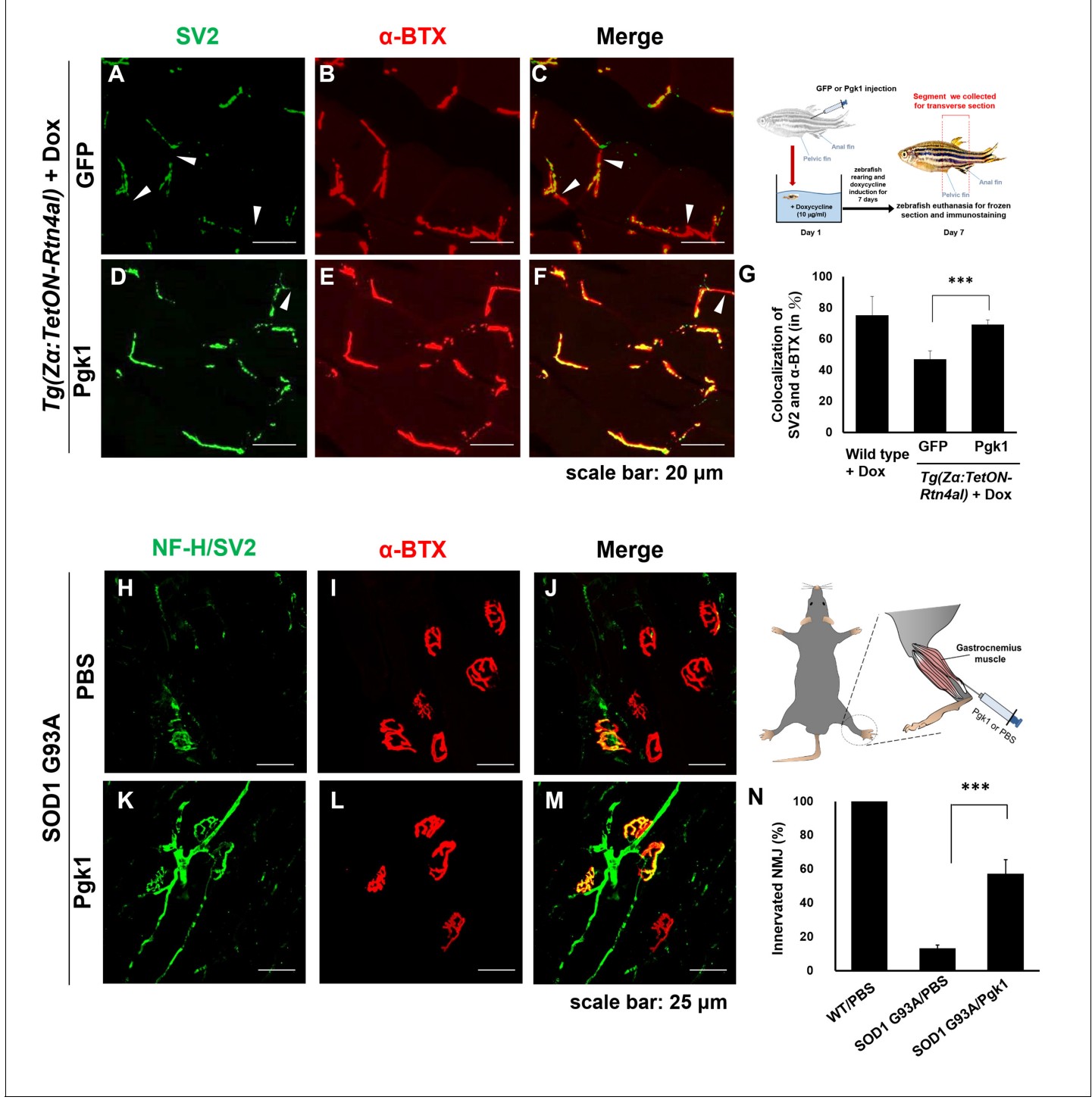

**Figure 6.** Supplementary addition of Pgk1 rescues denervation caused by NogoA-overexpression in the muscle cells of zebrafish, as well as ALS mouse model. NMJ phenotype of transgenic zebrafish *Tg(Zα:TetON-Rtn4al)* harboring Rtn4al (NogoA homolog) cDNA driven by a Dox-inducible muscle-specific α-actin promoter after intramuscular injection of (**A–C**) GFP protein control and (**D–F**) Pgk1. Axons were labeled by synaptic vesicle glycoprotein 2A (SV2) in green, while postsynaptic receptors were labeled by α-Bungarotoxin (α-BTX) in red. (**G**) Statistical analysis of the number of colocalized axons and postsynaptic receptors in muscle of NogoA-overexpression zebrafish ALS-like model using Student's *t*-test (***, p<0.001). NMJ phenotype of ALS mouse model harboring human SOD1 G93A after intramuscular injection of (**H–J**) PBS and (**K–M**) Pgk1 in the gastrocnemius muscle of the right hind leg. Neurofilament-H (NF-H) and SV2 labeled by green fluorescent signal were used to detect the axon terminal of motoneurons, while α-BTX labeled by red fluorescence signal was used to detect the acetylcholine receptor on motor endplates. (**N**) Statistical analysis of the number of

*Figure 6 continued on next page*

*Figure 6 continued*

innervated NMJ among PBS-injected WT mouse, PBS-injected SOD1 G93A mouse and Pgk1-injected SOD1 G93A mouse. Statistical analysis used Student's *t*-test (***, significant difference at p<0.001).

DOI: https://doi.org/10.7554/eLife.49175.016

The following video and figure supplements are available for figure 6:

**Figure supplement 1.** Overexpression of Rtn4al in adult zebrafish muscle causes NMJ denervation.
DOI: https://doi.org/10.7554/eLife.49175.017

**Figure supplement 2.** Administration of Pgk1 could delay NMJ denervation caused by overexpression of Rtn4al in adult zebrafish muscle.
DOI: https://doi.org/10.7554/eLife.49175.018

**Figure supplement 3.** The establishment and motor neuron differentiation of iPSC derived from an ALS patient with SOD1 mutated at G85R.
DOI: https://doi.org/10.7554/eLife.49175.019

**Figure supplement 4.** Pgk1 reduces the protein level of p-Cofilin-S3 in NSC34-SOD1 G93A and human iPS-SOD1 G85R cells.
DOI: https://doi.org/10.7554/eLife.49175.020

**Figure supplement 5.** Effect of Pgk1 injection on the exercise capability of hind leg of ALS mice.
DOI: https://doi.org/10.7554/eLife.49175.021

**Figure 6—video 1.** Sample number 1: The SOD1 G93A mouse right hind leg gastrocnemius injection PBS.
DOI: https://doi.org/10.7554/eLife.49175.022

**Figure 6—video 2.** Sample number 2: The SOD1 G93A mouse right hind leg gastrocnemius injection PBS.
DOI: https://doi.org/10.7554/eLife.49175.023

**Figure 6—video 3.** Sample number 3: The SOD1 G93A mouse right hind leg gastrocnemius injection PBS.
DOI: https://doi.org/10.7554/eLife.49175.024

**Figure 6—video 4.** Sample number 4: The SOD1 G93A mouse right hind leg gastrocnemius injection Pgk1.
DOI: https://doi.org/10.7554/eLife.49175.025

**Figure 6—video 5.** Sample number 5: The SOD1 G93A mouse right hind leg gastrocnemius injection Pgk1.
DOI: https://doi.org/10.7554/eLife.49175.026

**Figure 6—video 6.** Sample number 6: The SOD1 G93A mouse right hind leg gastrocnemius injection Pgk1.
DOI: https://doi.org/10.7554/eLife.49175.027

Furthermore, we examined the muscle contraction ability of hind leg in 130-day-old mice. In the WT/PBS group, the muscle contraction of both hind legs was normal, exhibiting strong movement. In contrast, in the SOD1 G93A/PBS group, muscle contraction of both hind legs was extremely poor (*Figure 6—videos 1–3*; *Figure 6—figure supplement 5*). Nevertheless, in the SOD1 G93A/Pgk1 group, muscle contraction of the left hind leg was as poor as that of the SOD1 G93A/PBS mice. Interestingly, muscle contraction of Pgk1-injected right hind leg remained functional, exhibiting a superior movement (*Figure 6—videos 4–6*). Exercise capability value of the SOD1 G93A/Pgk1 group was significantly higher than that of the SOD1 G93A/PBS group (*Figure 6—figure supplement 5*). Additionally, we found that the proportion of innervated NMJ in the gastrocnemius muscle of the Pgk1-injected right hind leg was increased compared to that of the left hind leg. Taken together, it is suggested that the Pgk1-injected right hind leg of SOD1 G93A mice could maintain some normal motor neurons to innervate muscle contraction.

Our collective findings reveal a novel target against the inhibition of neuron maturation and NOM by overexpression of NogoA through a Nogo66-independent pathway. NogoA has been known to inhibit neuron regeneration *via* Nogo66 and NiGΔ20 domains. To date, only Nogo66-interacting receptors were found. In this study, we revealed that overexpression of NogoA in muscle inhibits NOM through decreasing Pgk1 secretion (ePgk1), but without inducing the canonical Nogo66/NgR pathway. Here, we demonstrated that the addition of Pgk1 to media could promote NOM by reducing p-Cofilin-S3 through decreasing the phosphorylation of the Pak1/P38/MK2/Limk1-S323 signaling pathway *via* decreased Rac1-GTP (*Figure 7*). Whether NogoA also affects the secretion of other trophic factors to inhibit neuronal maturation and maintain NMJ integrity remains a topic for further study.

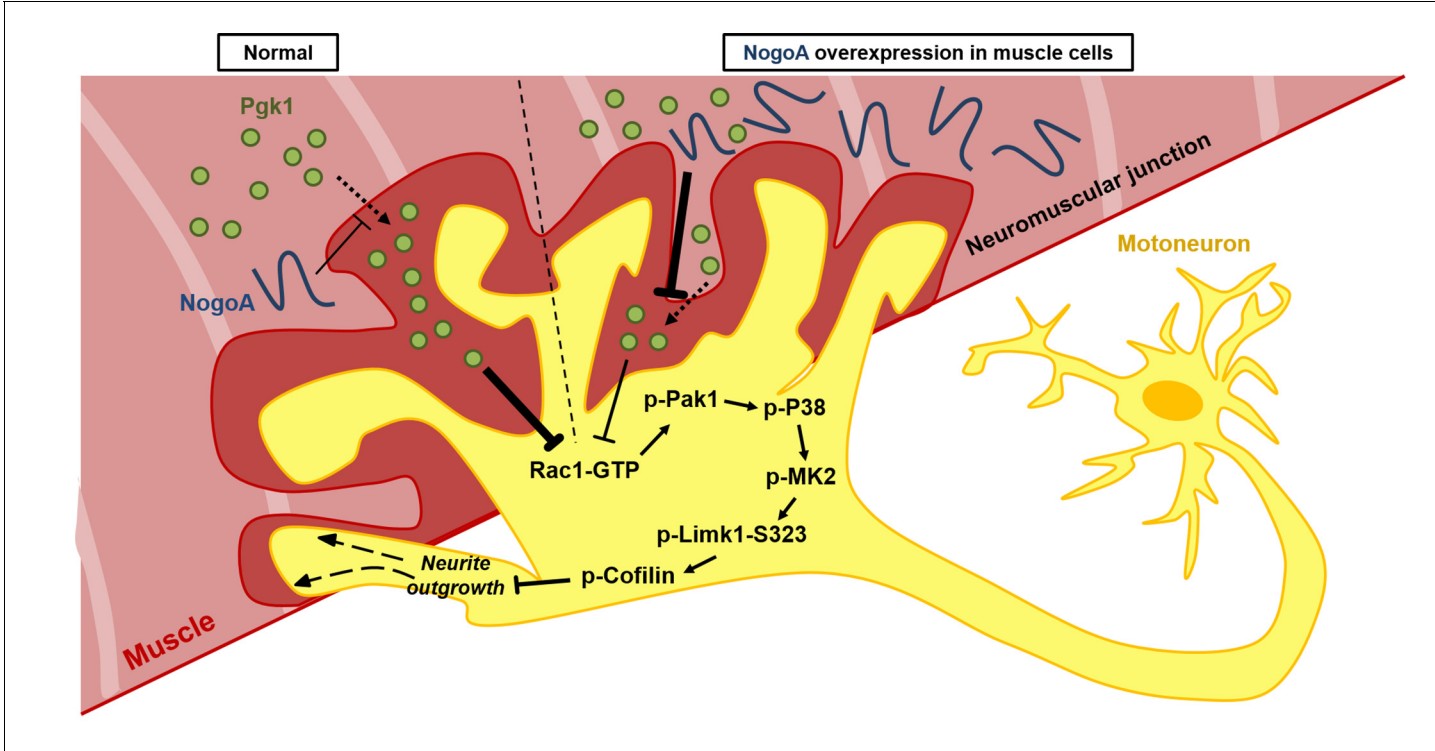

**Figure 7.** Diagrams used to illustrate the extracellular Pgk1-mediated signal pathway proposed by this study to demonstrate how Pgk1 secreted from NogoA-overexpressed muscle cells affects the neurite outgrowth of motor neurons. Left panel illustrates the small amount of NogoA presenting in muscle cells slightly reduces the amount of Pgk1 secreted from these muscle cells, resulting in strong inhibition of the Rac1-GTP/p-Pak1-T423/p-P38-T180/p-MK2-T334/p-Limk1-S323 axis within motor neurons, which, in turn, decreases the degree of p-Cofilin-S3. Consequently, the neurite outgrowth of motor neurons is developed. Right panel illustrates that overexpression of NogoA in muscle cells greatly reduces the amount of Pgk1 secreted from these muscle cells, resulting in weak inhibition of the Rac1-GTP/p-Pak1-T423/p-P38-T180/p-MK2-T334/p-Limk1-S323 axis within motor neurons, which, in turn, increases the degree of p-Cofilin-S3. Consequently, the neurite outgrowth of motor neurons is inhibited.
DOI: https://doi.org/10.7554/eLife.49175.028

## Discussion

### Overexpression of NogoA in cells affects the secretion of proteins

The expression of NogoA can generally affect the secretion of proteins. For example, the amount of insulin secreted from beta cells is increased in NogoA-deficient mice (*Bonal et al., 2013*). In PC12 cells, *Zhong et al. (2015)* demonstrated that silencing NogoA results in reducing the secretion of inflammatory factors, such as tumor necrosis factor-alpha and interleukin-6 . On the other hand, overexpression of NogoA increases $A\beta_{42}$ secretion in rat cerebral cortical neurons (*Xiao et al., 2012*). In this study, when NSC34 motoneuron cells were cultured in CM obtained from cultured Sol8 myoblast cells, which harbor a NogoA-overexpression plasmid, we found that NOM was compromised. Furthermore, we demonstrated that Pgk1 is a key factor contributing to the locomotive properties of neurons in terms of NOM, neurite-bearing cell number and the decrease of p-Cofilin in neuron cells. This finding further demonstrates that improper crosstalk between muscle and nerve tissues impedes the NOM. Although Pgk1 addition could rescue failure NOM induced by culturing NSC34 cells in Sol8-NogoA CM, we noticed that the efficacy of enhancement mediated by Pgk1 alone could barely reach levels as high as those of NSC34 cells cultured in CM from Sol8-vector, suggesting that Pgk1 may not be the only secreted protein involved in enhancing synaptic growth.

### Enhancement of NOM is a noncanonical function of ePgk1

To address the reduced secretion of Pgk1 from muscle cells by NogoA, as well as inhibition of the neuronal intracellular Rac1 pathway by ePgk1, we turn to the literature. First, *Lay et al. (2000)* found that Pgk1 secreted from tumor cells exhibits disulfide reductase activity, resulting in the release of

angiostatin through reducing the disulfide bond of plasmin. *Wang et al. (2007)* also reported that CXCR4 could positively regulate the expression and secretion of Pgk1 in cancer cells. Overexpression of Pgk1 causes an increase of angiostatin, which, in turn, reduces the secretion of vascular endothelial growth factor and interleukin-8. At metastatic sites, a high level of CXCL12 reduces Pgk1 expression, resulting in reduced angiostatin function, which activates angiogenesis. Interestingly, when Rtn4a/NogoA is overexpressed in the somite boundary cells of zebrafish embryos, *Lin et al. (2017)* found that the expression of Cxcr4a is decreased. *Aase et al. (2007)* and *Yi et al. (2011)* reported that angiomotin (Amot), a cell membrane receptor of angiostatin, negatively controls Rac1 activity in endothelial and epithelial cells. Based on this convincing evidence from the literature, we speculate that NogoA/Rtn4a-overexpressed muscle cells may cause a decrease of Cxcr4a expression, resulting in reducing of Pgk1 secretion. The reduced ePgk1 may then cause a decrease in extracellular proteins, such as angiostatin, which, in turn, would increase the activity of Rac1, finally causing a physiological microenvironment for motor neurons that would not favor neurite outgrowth.

*Boyd et al. (2017)* reported the intracellular, bioenergetic function of Pgk1 in neuron cells, while we described the effect of Pgk1 secreted from muscle cells on NOM through a bioenergetic- and Nogo66-independent pathway. As *Boyd et al. (2017)* reported, when Pgk1 is insufficiently expressed, its participation in the glycolysis pathway is not effective, resulting in a reduced production of ATP, which, in turn, causes impaired neurological function. Thus, they concluded that Pgk1 is expressed in motor neurons, but also has a bioenergetics function. However, since the effect of *pgk1*-MO and *pgk1* mRNA was not studied in the context of expression in muscle tissue, it remains unclear if the motor neuron phenotype results from the direct effect of Pgk1 within neurons or from indirect effect by defective function of surrounding cells or tissues. Therefore, in this study, we employed a muscle-specific promoter combined with the CRISPR/Cas9 system to specifically knock down Pgk1 in muscle cells, and we found that NOM was inhibited in adjacent nerve cells. Interestingly, when we used the muscle promoter to specifically overexpress Pgk1 in muscle cells, the failed NOM induced by NogoA/Rtn4al overexpression in muscle cells could be rescued. Additionally, we demonstrated that addition of Pgk1 in CM and DM had no impact on the efficiency of intracellular glycolysis in cultured NSC34 cells. In fact, the addition of Pgk1 enables the enhancement of NOM in NSC34 cells, even under reduced glycolytic condition. Additionally, when Pgk1 was ubiquitously knocked down, as *Boyd et al. (2017)* reported, the total amount of Pgk1 was reduced, including the small amount of secreted Pgk1 from muscle cells, resulting in a phenotype similar to what we observed in this study. Based on this line of evidence, we demonstrated that ePgk1 enhances NOM in a manner functionally independent of its intracellular, canonical role as a supplier of energy, as described by *Boyd et al. (2017)*.

## Potential application of ePgk1

Thus, we have identified a Pgk1/p-Limk1-S323/p-Cofilin-S3 axis in which secreted Pgk1 emerges as an extracellular trigger in the NOM. As illustrated in *Figure 6*, the following cascade of events takes place. ePgk1 decreases p-Limk1 at S323, which, in turn, results in the decrease of p-Cofilin-S3 and, finally, the rescue of NOM inhibited by culturing in Sol8-NogoA CM. In the process of investigating this novel signaling pathway, we found that overexpression of NogoA in muscle cells resulted in altering the amount of some secreted proteins in muscle cells. For example, the amount of secreted Pgk1 is reduced, which, in turn, inhibits the NOM. Meanwhile, we know that NogoA, as marker of morbidity, overexpresses in the muscle cells of ALS patients at an early stage (*Pradat et al., 2007*). Because NogoA does not contain a signal peptide (*Chen et al., 2000*), it is barely detectable in patients' blood (*Harel et al., 2009*). Therefore, we have detected a property of Pgk1 in serum that may serve as an indicator of incipient ALS. More recently, *Meininger et al. (2017)* reported that treatment with Ozanezumab, a humanized monoclonal antibody against NogoA, showed no significant improvement for ALS patients in phase two trial, suggesting that NogoA may not be an effective therapeutic target. Moving away from this target, we proposed in the present study that secreted protein from NogoA-overexpressed muscle cells is a highly promising alternative therapeutic target. More specifically, we showed that the reduced amount of secreted Pgk1 causes neurodegeneration and that the restorative effect of Pgk1 on NOM does not arise through the Nogo66/NgR/ROCK/Limk1-T508 pathway. Although intercellular Nogo66/NgR interaction is blocked by NogoA antibody, it may not change the fact that NogoA is still overexpressed within muscle cells,

resulting in the decrease of Pgk1 secretion. Meanwhile, Pgk1 deficiency is inherited as an X-linked recessive genetic trait. It is caused by mutated Pgk1, the folding and stability of which are altered (*Chiarelli et al., 2012*; *Pey et al., 2014*). Pgk1 deficiency is accompanied by suffering from muscle lesions, motor neuron defect, neurological dysfunction and myopathy (*Valentini et al., 2013*; *Matsumaru et al., 2017*), as well as susceptibility to Parkinson's disease (*Sakaue et al., 2017*). Therefore, it can be reasonably speculated that the abnormal secretion of Pgk1 may affect neuronal development, which supports our finding that Pgk1 secreted from muscle cells plays a novel role in NOM and neuronal development.

## Materials and methods

### Establishment of stable cell lines harboring Sol8-vector and Sol8-NogoA
The lentivirus vectors pAS4.1w.Pbsd-aOn and pAS4.1w.Ppuro-aOn, purchased from the National RNAi Core Facility Platform, Academia Sinica, Taiwan, were used to establish Sol8 myoblasts (RRID: CVCL_6449), which express transgene conditionally. The Sol8-vector was a plasmid pAS4.1w.Ppuro-aOn, while Sol8-NogoA was a plasmid pAS4.1w.Ppuro-aOn with an insert of *NogoA* cDNA, pAS4.1w.Ppuro-aOn-NogoA. Expression plasmids pCMVΔR8.91 and pMD.G, and lentivirus vector were transfected into cells by Lipofectamine 2000 (Invitrogen) with a 0.9:0.1:1 ratio, respectively, according to the manufacturer's protocol. The culture medium containing infected virus particles of HEK293T cells (RRID:CVCL_0063, confirmed negative for mycoplasma contamination) was collected at 24 and 48 hr post-transfection after spinning down at 1250 rpm at room temperature. When Sol8 cells grew up to 40% confluency, the supernatant was saved and transferred to Sol8 myoblasts with addition of Polybrene (Sigma) at the concentration of 8 μg/ml for 24 hr. After incubation, the medium was replaced by a fresh growth medium containing Puromycin at the concentration of 4 μg/ ml.

### NSC34 cells were cultured in conditioned medium
After cells reached at 90% confluency, medium was replaced by horse serum (HS)-containing differentiation medium DMEM (2.5% HS, 100 units/ml Penicillin and 0.1 mg Streptomycin) containing Dox at 1 μg/ml. After 24 hr incubation, medium was again replaced by FBS-containing DMEM containing Dox. After another 24 hr incubation, the conditioned medium of Sol8 myoblasts was collected and centrifuged at 1000 rpm. The saved supernatant was used to culture NSC34 cells (*Cashman et al., 1992*) until they reached at 60% confluency. Medium was then replaced by fresh FBS-containing DMEM containing Dox and continuously cultured for 24 hr. The conditioned media of cultured Sol8 myoblasts was collected by centrifugation and used to culture NSC34 cells for 48 hr. Then, cell lysate was obtained for Western blot analysis.

### *In vitro* platform of NOM derived from NSC34 cells
NSC34 cells were seeded in 10% FBS DMEM medium onto six-well plates at cell density of $3 \times 10^4$ per well. After 24 hr incubation, culture medium was replaced by 2.5% FBS DMEM medium to stimulate differentiation of NSC34. After another 24 hr incubation, medium was exchanged for Sol8-conditioned medium and incubated for 96 hr. Finally, cells were fixed with 4% PFA overnight. Neurites extending from each cell were captured by microscopy (Leica) under 200 times magnification and labeled by Neurolucida Neuron Tracing Software 9.0 (Neurolucida 9.0). For each image, 40 cells cultured in either CM or DM were randomly chosen to measure the length of neurites. Data were obtained from three independent trials (120 cells in total). To profile neurite length distribution, we grouped the neurite length into several groups, namely every 25 intervals under 50 μm, every 50 intervals under 200 μm, and 201–300 and 301–600 μm, and calculated the number of cells in each range. To count the number of neurite-bearing cells, we calculated the number of cells with neurites at least 50 μm in length in each range.

### Frozen section and immunostaining of zebrafish, mice and NSC34 cells
The procedures of frozen section and immunostaining of zebrafish and mice tissues and cells were previously described by *Lin et al. (2019)*, except primary antibodies, including anti-Syn1 (RRID:AB_2200097; 1:250), anti-neurofilament H (NF-H) (RRID:AB_91202; 1:500) and anti-acetyl-tubulin (RRID:

AB_297928; 1:500), and secondary antibodies, including anti-rabbit Cy2 (RRID:AB_827264; 1:200) and Alexa Fluor 488-conjugated Goat anti-mouse IgG (RRID:AB_2576208; 1:500), were used. The patterns were observed under a Laser Scanning Confocal Fluorescence Microscope (Zeiss). Nuclei were counterstained with 4',6-diamidino-2-phenylindole (DAPI) (Sigma).

## Two-dimensional (2D) SDS-PAGE and LC/MS-MS analyses

Total proteins of CM from cultured muscle cells in Sol8-NogoA and Sol8-vector were separately collected. Before total proteins were analyzed on 2D SDS-PAGE, Cibacron Blue F3G-A and ammonium sulfate precipitation were performed to remove albumin contained in CM (*Colantonio et al., 2005*), while remaining proteins were mixed with acetone, precipitated by centrifugation, and dissolved in rehydration buffer (8 M urea, 2% NP-40, and 100 mM DTT). The strips were subjected to rehydration overnight and isoelectric focusing. After isoelectric focusing, the strips were removed and equilibrated. The equilibrated strips were run on the gel as previously described by *Sanchez et al. (1999)*. Gels were stained by Coomassie blue and dried between two sheets of cellophane. Protein spots were excised and subjected to in-gel digestion. The procedures of in-gel digestion and LC/MS-MS analyses were done as previously described in *Fu et al. (2012)*. First, we employed 2D electrophoresis to analyze total protein content in CM. Next, we selected a total of 20 spots at random, all displaying more or less intensity, and we then made a comparison between the Sol8-NogoA CM protein pattern and that of Sol8-vector CM. Following LC MS/MS, 42 candidate proteins were identified (see *Figure 1—figure supplement 2*). We picked up 10 proteins reduced in Sol8-NogoA CM, cloned their cDNAs, and performed genetic engineering to overexpress these genes individually in Sol8-NogoA cells. We also picked up 10 proteins enhanced in Sol8-NogoA CM, determined their cDNAs, and designed the corresponding siRNAs to knock down their expression individually in Sol8-NogoA cells. All corresponding CMs were collected to culture NSC34 cells separately. After 48 hr in culture, the degree of p-Cofilin expression in NSC34 cells was quantified, and the genes exhibiting lower expression of p-Cofilin were further selected.

## Zebrafish husbandry and microscopy observation

Zebrafish (*Danio rerio*) wild-type AB strain (RRID:ZIRC_ZL1) and transgenic lines *Tg(mnx1:GFP)* (RRID:ZIRC_ZL1163) (*Flanagan-Steet et al., 2005*), purchased from ZIRC, and *Tg(Zα:TetON-Rtn4al)* (*Lin et al., 2019*) were used. Production and stage identification of embryos followed the description by *Lin et al. (2019)*.

## Ethics statement

The Mackay Medical College Institutional Animal Care and Use Committee (IACUC) reviewed and approved the protocol described below (MMC-A1060009).

## Using CRISPR/Cas9 system to perform knockdown experiments

The sgRNAs of *pgk1* (TGGACGTGAAAGGAAAGC) and *pgam2* (CCTGGAGGAGGCGAAACG) were subcloned into plasmid pDR274 and transcribed into mRNAs *in vitro* by the mMESSAGE mMACHINE T7 ULTRA kit (Life Technologies). The linearized plasmid pZα-Cas9, in which Cas9 was driven by the muscle-specific alpha-actin promoter of zebrafish, was coinjected separately with *pgk1* and *pgam2* (control) sgRNAs into zebrafish embryos, and phenotypes were observed at 30 hpf.

## Plasmid construction

The coding region of *Limk1* cDNA (NM_010717.3) fused with FLAG-tag was inserted into pCS2$^+$ to generate pCS2-Limk1-flag. Mutated forms of Limk1, such as Limk1 S323A and T508V, were obtained by PCR and inserted into pCS2$^+$ to generate pCS2-Limk1-S323A-flag and pCS2-Limk1-T508V-flag, respectively. Similarly, *Pak1* cDNA (NM_011035.2) with FLAG-tag was inserted into pcDNA3 to generate pcDNA3-Pak1-flag. Additionally, zebrafish *pgk1* (NM_213387.1) fused with P2A-RFP and Cas9 fused with P2A-RFP were synthesized and engineered with zebrafish muscle-specific α-actin promoter to generate pZα-Pgk1 and pZα-Cas9 for zebrafish embryo microinjection.

## Fluorescence-activated cell sorting (FACS)

Procedure of the dissociation of zebrafish embryonic cells was followed the previous study reported by *Lin et al. (2017)* with some modifications. Briefly, the WT embryos and embryos injected with pZα-Cas9, pZα-Cas9 plus *pgk1* sgRNA, pZα-Cas9 plus *pgam2* sgRNA and pZα-Pgk1 at 30 hpf were incubated with trypsin (Sigma) for 20 min at room temperature. After treatment, embryos were shattered by gently pipetting, followed by separating cells completely from tissues. Then, the RFP-expressing cells were sorted and collected by Cell Sorters (BD FACSAria III).

## Protein expression and purification

Protein purification followed the procedures described by *Fu et al. (2017)*. Plasmids expressing FLAG-fusion proteins were transfected into the HEK293T cell line. Transfected cells were lysed using Pierce IP lysis buffer (ThermoFisher Scientific; TFS) with protease inhibitor cocktail (Roche). After cell debris was removed by centrifugation, anti-FLAG M2 affinity gel beads were added to cell extracts and incubated for 16 hr at 4°C. Beads-FLAG-protein complex was eluted by incubation with 3X FLAG peptide for one hr. The FLAG-fusion proteins eluate was restored and used for the following experiment. Plasmids pGEX-GST and pGEX-GST-Nogo66 were used to express recombinant proteins using 0.1 mM Isopropyl β-D-1-thiogalactopyranoside induction for 1 hr at 37°C in an *Escherichia coli* BL21 (ATCC BAA-1025TM) expression system and purified by Glutathione resin (Clontech).

## Western blot analysis

Total proteins extracted from embryos were analyzed on a 10% SDS-PAGE followed by Western blot analysis according to the procedures described by *Lin et al. (2017)*. The antibodies against NogoA (RRID:AB_650319; 1:500), Cofilin (RRID:AB_10622000; 1:1000), phosphorylated Cofilin at S3 (RRID:AB_2080597; 1:1000), Rho-associated protein kinase 2 (ROCK2) (RRID:AB_10829468; 1:2000), phosphorylated ROCK2 at Y256 (RRID:AB_2182301; 1:2000), Epidermal growth factor receptor (EGFR) (RRID:AB_2246311; 1:500), phosphorylated EGFR at Y1173 (RRID:AB_331795; 1:2000), Akt (Protein kinase B; PKB) (RRID:AB_329827; 1:1000), phosphorylated Akt at S473 (RRID:AB_2315049; 1:1000), LIM domain kinase 1 (Limk1) (RRID:AB_648350; 1:500), phosphorylated Limk1 at S323 (RRID:AB_2136940; 1:1000), phosphorylated Limk1 at T508 (RRID:AB_2136943; 1:500), Phosphoglycerate kinase 1 (Pgk1) (RRID:AB_2161220; 1:2000) (RRID:AB_2268000 for zebrafish; 1:1000), Phosphoglycerate mutase 2 (Pgam2) (RRID:AB_1951200; 1:1000), Ras-related C3 botulinum toxin substrate 1 (Rac1) (RRID:AB_2721173; 1:500), p21-activated kinase 1 (Pak1) (RRID:AB_330222; 1:1000), phosphorylated Pak1 at T423 (RRID:AB_330220; 1:1000), P38 mitogen-activated protein kinases (P38) (RRID:AB_330713; 1:1000), phosphorylated P38 at T180 (RRID:AB_331641; 1:1000), MAP kinase-activated protein kinase 2 (MK2) (RRID:AB_10694238; 1:1000), phosphorylated MK2 at T334 (RRID:AB_490936; 1:1000), Growth Associated Protein 43 (GAP43) (RRID:AB_443303; 1:1000), Choline acetyltransferase (ChAT) (RRID:AB_2244867; 1:1000), Microtubule-associated protein 2 (MAP2) (RRID:AB_2138153; 1:1000), α-tubulin (RRID:AB_477579; 1:5000), Myc (RRID:AB_439680; 1:2000), Flag (RRID:AB_446355; 1:5000), rabbit anti-sheep-HRP (RRID:AB_656968; 1:5000), goat anti-mouse-HRP (RRID:AB_955439; 1:5000) and goat anti-rabbit-HRP (RRID:AB_631746; 1:5000) were used.

## Glycolysis stress test assay

Seahorse XF glycolysis stress test kit (Seahorse Bioscience, USA) was used to measure the extracellular acidification rate (ECAR; mpH/min), also called the $H^+$ production rate, of NSC34 cells. Briefly, NSC34 cells were seeded in a 10% FBS DMEM medium onto SeaHorse XF 24-well plates at a cell density of $2.5 \times 10^5$ per well. After 24 hr incubation, culture medium was replaced by a 2.5% FBS DMEM medium to initiate cell differentiation. Then, NSC34 cells were treated with either Pgk1 (Bio-Techne R and D Systems; BT) at 33 ng/ml or 3PO (SA) at 3 M for two days. When medium was replaced by glucose-free Seahorse 24-well XF Cell Culture, cells were continuously incubated in a non-$CO_2$ incubator at 37°C for 30 min before assay. To measure the ECAR of the surrounding media, glycolytic flux, such as basal glycolysis, glycolytic capacity and glycolytic reserve, was analyzed by the sequential addition of 10 mM glucose, 1 M Oligomycin and 50 mM 2-deoxyglucose in an XF24 flux analyzer.

## Medium for culturing iPS cell lines

The generation of healthy- and ALS-iPS cells from peripheral blood mononuclear cells of hSOD1 G85R was approved under the Institutional Review Board of Hualien Tzu Chi General Hospital, Hualien, Taiwan (IRB-105–131-A). Briefly, iPS cells were established with Cytotune-ips 2.0 Sendai Reprogramming kit (TFS) and cultured in Essential eight medium (TFS) on 1.0% hES qualified Matrigel (Becton Dickinson)-coated cell culture dishes. The cells were passaged per 3–5 days using Accutase (Merck-Millipore), mechanically scraped, and then reseeded at a 1:5 to 1:10 ratio. The culture medium was refreshed daily.

## Motor neuron induction and maturation

When cell confluence was reached at 80% on the culture dish, cells were treated with Accutase for 2–5 min and scraped. The cell clumps were dissociated into 200–300 mm clusters and transferred to noncoated Petri dishes for 48 hr for embryoid body (EB) formation. The EBs were applied by the modified BiSF neural induction method, according to the methods described previously (*Chen et al., 2015*). The differentiation medium within the first two days was Essential 6 (TFS). Subsequently, on D2 of differentiation, the cell culture medium was changed to DMEM-F12 supplied with 1% N2 supplement (TFS), 1 mM NEAAs and 2 mM glutamate. Neuron-inducing factors, including 3 µM CHIR99021 (SA), 10 µM SB431542 (SA) and 10 ng/ml recombinant human FGF-2 (R and D Systems) were added on D2 for 2 days. On D4, the neural induction medium was removed, and the neurospheres were cultured in neurobasal medium (TFS) with 1% N2 supplement. Motor neuron patterning factors including 0.5 µM retinoic acid (RA, SA), 0.5 µM purmorphamine and 0.5 µM Smoothened Agonist (SAG, SA) were added from D4 for 5 days, and then reduced dosage of motor neuron patterning factors (0.1 µM RA and 0.25 µM SAG) were added for another 5 days. After complete motor neuron patterning, the cells were dissociated into small clumps by Accutase and seeded on 1% Matrigel (TFS)-coated cell culture dishes for neural maturation. The motor neuron progenitors (MNPs) proliferated in neurobasal medium with 2% B27 supplement (TFS), 10 ng/mL brain-derived neurotrophic factor (BT), 10 ng/ml Glial Cell Line-derived Neurotrophic Factor (BT) and 1 µM Dibutyryl-cAMP (dbcAMP, SA). During the passaging of MNPs with Accutase, providing 10 µM Y27632 (SA) or RevitaCell Supplement (TFS) effectively attenuated dissociation-triggered cell death.

## Intramuscular injection of Pgk1

We directly injected Pgk1 (2 µg) into muscle of adult fish from transgenic line *Tg(Zα:TetON-Rtn4al)*, followed by Dox immersion for one week when fish exhibited motorneuron degeneration. The percentage of overlapping signals between SV2 and α-BTX were calculated. We also performed intramuscular injection of Pgk1 into the gastrocnemius muscle of the right hind leg of six 60-day-old (at P60) transgenic SOD1 G93A mice (RRID:IMSR_JAX:002726). Each time, a volume of 80 µl Pgk1 dissolved in PBS solution in a concentration of 375 µg/ml was injected. On three of these frozen longitudinal sections, we performed immunofluorescence staining to quantify the proportion of innervated NMJ in the gastrocnemius muscle of the right hind leg at P75. The remaining three were continuously given Pgk1 injections every 15 days until mice were 120 days old. Six (three for NMJ and three for exercise studies) wild-type mice injected with 80 µl PBS (WT/PBS) into the right leg served as the sham group, and six (three for NMJ and three for exercise studies) SOD1 G93A mice injected with PBS (SOD1 G93A/PBS) served as the negative control group. When the SOD1 G93A/PBS and SOD1 G93A/Pgk1 mice were 130 days old, their movement was recorded by video, and exercise capability of both hind legs was evaluated by quantifying the number of leg contractions within one min. Two legs of each mouse were counted separately. The exercise capability value of each mouse was calculated from the increased fold(s) in the number of contractions between injected right leg versus untreated left leg. Data of each group were averaged from three mouse samples and presented as mean ±S.D.

## Acknowledgements

This work was supported by Prof. Jin-Chuan Sheu, Dr. Hsiao-Ching Nien and Mr. Spencer Lee, the Liver Disease Prevention and Treatment Research Foundation, Taiwan. We thank TC3 and TC5 Technology Commons, College of Life Science, NTU, for providing the FACS, Microscope and 2D PAGE

equipments. We also thank Biomedical Instrumental Center, Mackay Medical College, for providing the SeaHorse XF 24 for Glycolysis Stress Test Assay. We also thank Prof. Hsinyu Lee, College of Life Science, NTU, for providing plasmids pCMVΔR8.91 and pMD.G. We also thank Dr. Neil Cashman, University of Toronto, for providing NSC34 cell line.

## Additional information

### Funding

| Funder | Grant reference number | Author |
|--------|------------------------|--------|
| Ministry of Science and Technology, Taiwan | 107-2311-B-715-001 | Huai Jen Tsai |
| Ministry of Science and Technology, Taiwan | 107–2314-B-303-003 | Shinn Zong Lin |
| Liver Disease Prevention and Treatment Research Foundation, Taiwan | | Cheng Yung Lin Huai Jen Tsai |

The funders had no role in study design, data collection and interpretation, or the decision to submit the work for publication.

### Author contributions

Cheng Yung Lin, Data curation, Formal analysis, Investigation, Writing—original draft; Chia Lun Wu, Data curation, Formal analysis, Investigation, Methodology, Writing—original draft; Kok Zhi Lee, Data curation, Investigation, Methodology, Writing—original draft; You Jei Chen, Po Hsiang Zhang, Data curation, Investigation, Methodology; Chia Yu Chang, Data curation, Methodology; Horng Jyh Harn, Resources, Methodology; Shinn Zong Lin, Resources, Funding acquisition; Huai Jen Tsai, Formal analysis, Investigation, Resources, Supervision, Funding acquisition, Writing—review and editing

### Author ORCIDs

Cheng Yung Lin (iD) https://orcid.org/0000-0001-9570-0218
Po Hsiang Zhang (iD) https://orcid.org/0000-0002-9745-0213
Huai Jen Tsai (iD) https://orcid.org/0000-0001-8242-4939

### Ethics

Animal experimentation: The Mackay Medical College Institutional Animal Care and Use Committee (IACUC) reviewed and approved the protocol described below (MMC-A1060009).

### Decision letter and Author response

Decision letter https://doi.org/10.7554/eLife.49175.030
Author response https://doi.org/10.7554/eLife.49175.031

## Additional files

### Data availability

All data generated or analysed during this study are included in the manuscript and supporting files.

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
