## [Decision Letter]

Thank you for submitting your work entitled "Extracellular Pgk1 enhances neurite outgrowth of motoneurons through Nogo66-independent targeting of NogoA" for consideration by *eLife*. Your article has been reviewed by three peer reviewers, including Fadel Tissir as the Reviewing Editor and Reviewer #1, and the evaluation has been overseen by a Senior Editor.

While the three reviewers found the work interesting, there are issues that need to be addressed by additional experiments. These include the validation and appropriate controls for Pgk1 Crispr/Cas experiments (two reviewers), the confirmation that overexpression of NogoA affects Pgk1 secretion in vivo (all reviewers), and the claim that Pgk1 function is NgR independent (two reviewers).

Addressing these points will require more than the typical 2 months for a standard *eLife* revision. We would be happy to re-consider the manuscript should you consider to revise along the suggestions of reviewers appended below.

*Reviewer #1:*

In this manuscript, Chen Yung Lin, Huai Jen Tsai and colleagues report that Pgk1 promotes neurites outgrowth of motor neurons independently of its glycolytic function and of Nogo66. They used downregulation, upregulation, and rescue experiments in cells (myoblasts, and NSC34 derived motor neuron cell-like cells), as well as zebrafish and mouse; and showed that:

1) Conditioned medium of myoblast cells overexpressing NogoA inhibits motor neuron neurites outgrowth

2) Analysis of the conditioned medium by 2D electrophoresis and LC MS shows that overexpression of NogoA reduces the level of secreted Pgk1.

3) Addition of Pgk1 restores neurites growth, decreases the amount of p-cofilin via pLimk-S323, and promotes neuronal differentiation and maturation.

4) Pgk1 blocking antibody has the opposite effect

5) The effect of Pgk1 on neurites growth in independent of its glycolytic role as Pgk1 mutants that have no catalytic activity are still capable to reduce p-cofilin levels and induce changes in neurites growth

6) CrispR/Cas-mediated inactivation of Pgk1 in fish muscle cells inhibits neurites outgrowth while upregulation of Pgk1 promotes (ectopic) growth.

7) Intramuscular injection of Pgk1 recues the neuromuscular junction defects in Rtn4al transgenic Fish and SOD1 transgenic mice.

Overall, the work is of high quality. The results are compelling, and virtually all possible controls have been done.

Points to address:

1) The text contains too many typographical and grammatical errors that make the manuscript difficult to understand sometimes. The text has to be edited to facilitate the reading and clarify the message.

2) Were all the "spots" identified by 2D electrophoresis subjected to MS analysis, and is Pgk1 the only protein identified by LC-MS? If not, why this one has been selected?

3) The validation of Cripr/cas9-mediated inactivation/upregulation of Pgk1 in zebrafish was done by monitoring the expression of the RFP reporter. While changes in RFP expression are obvious, this does not tell much about Pgk1 levels in the KO and transgenic Fish compared to endogenous expression. Western blot analysis would be helpful. Also, could the authors rule out any off-target effect as injection Cas9 alone (driven by the muscle specific α actin promoter, but in absence guide RNA) seems to produce some effects on motor axons (compare D-D" to C-C")?

*Reviewer #2:*

In the manuscript by Lin, et al., the authors identifiy Pgk1 as a novel secreted molecule that can stimulate neurite outgrowth. They further show that the secretion of Pgk1 is reduced when NogoA is expressed in the secreting cells, identify a potential mechanism by which NogoA reduces axon outgrowth independent of its interactions with Nogo receptors.

While the identification of this novel role for Pgk1 is very interesting, there are a number of issues throughout the study that are of major concern, including a lack of important controls. In particular are the following points:

1) Throughout the paper, there is inadequate information about "N"s in experiments, and inappropriate statistical analysis for multiple comparisons.

2) There is no indication of Pgk1 dose or timecourse throughout the entire study. Is the dose used physiologically relevant? Is there a dose dependent response?

3) Does the NogoA-Pgk1 secretion mechanism also function in primary muscle cells and motor neurons in culture? Is there evidence for NogoA over-expression inducing a reduction in Pgk1 expression in zebrafish and in mouse in vivo?

4) At the end of the Results section, the authors mention anecdotally that there is a motor behavior phenotype and provide videos. However, there is no quantification and no description in the Materials and methods section to indicate any type of quantitative assessment of motor behavior. How many mice were used in these experiments?

5) The authors conclude that Pgk1 functions independent of NgR. However, their only evidence for this is the lack of activation of downstream signaling cascades, and there are no positive controls included in these experiments. This is very indirect, and does not allow the authors to conclude Pgk1 function is independent of NgR.

6) How NogoA reduces the secretion of Pgk1 from muscle cells and how extracellular Pgk1 inhibits the intracellular Rac1-PAK pathway is unclear and needs additional investigation and/or discussion.

Additional comments on the specific Figures:

Figure 1:

The authors need to confirm that there is no NOGO-A in the conditioned media

The specificity of the Pgk1 Ab used in Figure 1B needs to be confirmed

Figure 2:

It is unclear how well the markers used (MAP2, ChAT, GAP43, SYN1) can actually reflect a differentiated status in NSC34 cells. These cells have very small processes, and therefore are unlikely to represent a truly differentiated state. In subsection “Supplementary addition of Pgk1 can enhance the maturation of NSC34 cells cultured in Sol8-NogoA CM”, the authors state "The number of neurons with Syn1 signal at the end of synapse could be restored to that cultured in Sol8-vector…". It is unlikely that these are synapses, they look more like growth cones and don't contact "postsynaptic" structures

Figure 4:

In zebrafish experiments, the authors use CRISPR/Cas9 system to knockout Pgk1 and Pgam2 from muscle cells. There needs to be data validating that sgRNAs targeting Pgk1 and Pgam2 are effective in reducing Pgk1 and Pgam2 expression, respectively. Similarly, in the muscle-specific over expression paradigm, the authors need to show that Pgk1 expression is increased.

While the authors provide information on the penetrance of the phenotypes, they do not quantify the extent of axonal extension in any meaningful way.

Figure 5:

The authors conclude that the effect of Pgk1 is NgR-independent due to the lack of p-ROCK2-Y256, p-EGFR-Y1173 and p-Akt-S473 (Figure 5—figure supplement 1). However, there is no positive control in this experiment. The authors should show that they are able to detect changes in these pathways in response to NgR activation by NOGO-A.

Figure 6:

Because the authors did not provide information on the dose-dependency and timecourse of Pgk1 in their in vitro experiments, it is unclear why they chose the specific dose and timing regimens for their in vivo experiments. Fish were injected with 2ug Pgk1, then placed in Dox+ water for 1 week. How long is the injected Pgk1 biologically active?

There is a discrepancy between what is in the Results section and the Materials and methods section with respect to the experiments done in mice in vivo. The authors indicate that mice were injected with Pkg1 once every 15 days between P60 and P120. However, in the Materials and methods section (546-548) they indicate tissue sections were taken at P75. Is the data presented in Figure 6 H-N from P75 or P120 mice?

The authors should provide images and other metrics (neurite outgrowth, expression of motor neuron specific genes) from their iPS-SOD G85R motor neuron cultures so the degree of differentiation these iPS cells can be assessed.

- Figure 6—figure supplement 2 panel B: Why is there no change in the levels of p-Cofilin in response to Pgk1 in the control cells? In Figure 1A, this same treatment causes a ~3-fold decrease p-Cofilin levels.

*Reviewer #3:*

In this manuscript, the authors found: 1) Overexpression of NogoA in a muscle cell line Sol8 reduced the secretion of Pgk1 detected from the condition medium; 2) Neurite outgrowth of NSC34 cells was inhibited by Sol8-NogoA CM (conditioned medium) and could be rescued by additional Pgk1; 3) Sol8-NogoA CM increased p-Cofilin S3, while extracellular Pgk1 decreased Rac1-GTP/p-Pak1-T423/p-P38-T180/p-MK2-T334/p-Limk1-S323/p-Cofilin-S3 pathway; 4) Conditional KO of Pgk1 in zebrafish muscle cells impaired motoneuron axon growth and the increase of Pgk1 in muscles cells resulted in ectopic growth of motoneuron axons; 5) Intramuscular injection of Pgk1 fostered motoneuon axon regrowth and NMJ formation in Rtn4al-overexpressing transgenic zebrafish and SOD1 transgenic mice. This study revealed a new indirect mechanism how NogoA in the muscle affected neurite growth of motoneurons. The finding is interesting and helpful for understanding the probable mechanisms in steering axon growth. However, there are some issues needed to be addressed.

1) Nogo-A is a transmembrane protein, it could exert its effects by direct interacting with its receptors on the neurons, or it can work indirectly through affecting other molecule expression or secretion in the muscle. The authors used CM from NogoA-overexpressing cells to treat neurons, which supports the indirect effect. Given NogoA is a transmembrane protein, it is not likely present in the CM. Without the co-culture experiment of muscle cells and motor neurons, it's inappropriate to exclude the possibility that NogoA in the muscle could also affects neurite growth through a direct ligand/receptor interaction. Therefore, the conclusion "In this study, we revealed that overexpression of NogoA in muscle inhibits NOM through decreasing Pgk1 secretion (ePgk1), but without inducing the canonical Nogo66/NgR pathway."(subsection “Intramuscular injection of Pgk1 is able to rescue NMJ denervation caused by Rtn4al-overexpression transgenic zebrafish and SOD1 G93A transgenic mice”) is not accurate.

2) Is there any evidence in vivo that NogoA upregulation in the muscle affected its Pgk1 production, i.e. the NogoA and Pgk1 levels in the SOD1 mutant mice or Rtn4al-overexpressing transgenic zebrafish? Or at least confirm whether overexpressing NogoA in the muscle of normal adult mice using viral vector will also affect Pgk1 secretion.

3) The authors seemed to confuse with neurite and synapse: subsection “in vitro platform of neurite outgrowth of motor neurons derived from NSC34 cells”; "Synapses extending from each cell were captured by microscope.…"; "calculate the length of synapses…"

4) In Figure 1—figure supplement 2, the authors need quantify the Pgk1 amount in the CM. In addition, there is no information about the source and the concentration of ePgk1 added into the culture. Whether is it comparable to the amount in the CM?

5) In the method of neurite outgrowth (subsection “in vitro platform of neurite outgrowth of motor neurons derived from NSC34 cells”), it says: "In each image, 40~50 400 cells were randomly chosen to calculate the length of synapses and the number of neurite-bearing cells, the synapses of which were at least 50 μm in length.". If only cells with neurite length longer than 50 μm were chosen, why the average neurite lengths in some groups of Figure 1E are shorter than 50 μm?

---

## [Author Response]

Reviewer #1:

Points to address:1) The text contains too many typographical and grammatical errors that make the manuscript difficult to understand sometimes. The text has to be edited to facilitate the reading and clarify the message.

I apologize for having made so many mistakes. I have asked an expert to edit this manuscript.

2) Were all the "spots" identified by 2D electrophoresis subjected to MS analysis, and is Pgk1 the only protein identified by LC-MS? If not, why this one has been selected?

In the revised Material and Methods section, we described in more detail how Pgk1 was selected for this study, as follows: “First, we employed 2D electrophoresis to analyze total protein content in CM. Next, we selected a total of 20 spots at random, all displaying more or less intensity, and we then made a comparison between the Sol8-NogoA CM protein pattern and that of Sol8-vector CM. Following LC MS/MS, 42 candidate proteins were identified (see Figure 1—figure supplement 2). We picked up 10 proteins reduced in Sol8-NogoA CM, cloned their cDNAs, and performed genetic engineering to overexpress these genes individually in Sol8-NogoA cells. We also picked up 10 proteins enhanced in Sol8-NogoA CM, determined their cDNAs, and designed the corresponding siRNAs to knock down their expression individually in Sol8-NogoA cells. All corresponding CMs were collected to culture NSC34 cells separately. After 48 hr in culture, the degree of p-Cofilin expression in NSC34 cells was quantified, and the genes exhibiting lower expression of p-Cofilin were further selected.”

The Results section has been revised as follows: “We further employed 2D electrophoresis, followed by LC MS/MS, to analyze the total protein content in CM. Among these examined proteins, we found that the level of Pgk1 protein contained in Sol8-NogoA CM was significantly reduced compared to that in Sol8-vector CM (see Figure 1—figure supplement 2). Furthermore, the degree of p-Cofilin expression in NSC34 cells cultured in CM from Pgk1-overexpressed Sol8-NogoA cells was significantly reduced compared with that in NSC34 cells cultured in CM from Sol8-NogoA cells. Therefore, Pgk1 was chosen for further study to confirm its potential role in NOM.”

3) The validation of Cripr/cas9-mediated inactivation/upregulation of Pgk1 in zebrafish was done by monitoring the expression of the RFP reporter. While changes in RFP expression are obvious, this does not tell much about Pgk1 levels in the KO and transgenic Fish compared to endogenous expression. Western blot analysis would be helpful.

In response to your suggestion, we added Crispr/cas9-mediated inactivation and upregulation of Pgk1 in zebrafish muscle cells and highlighted in blue the relevant changes in the Results section, as follows: “We used muscle-specific α-actin promoter to drive overexpression of Cas9 in muscle cells in order to exclusively knock out the Pgk1 gene. […] As suggested by quantification of the axonal extension phenotype shown in Figure 4—figure supplement 2, the increase of Pgk1 in muscle cells enhances NOM.”

Also, could the authors rule out any off-target effect as injection Cas9 alone (driven by the muscle specific α actin promoter, but in absence guide RNA) seems to produce some effects on motor axons (compare D-D" to C-C")?

In a previous manuscript, our image of the Cas9-injected embryo seemed to reflect delayed development. To confirm this result, we repeated this experiment and found that almost Cas9-injected embryos did not exhibit defective motoneurons except that only very low percentage of injected embryos exhibited delayed development. Therefore, to avoid misinterpretation, we have substituted a new image in the revised Figure 4D.

When we injected either Cas9 alone or pZα-Cas9, off-target effect might have have been occasionally found. For example, sometimes few injected embryos seemed to exhibit developmental delay. However, we noticed that this defect is commonly observed in zebrafish embryos injected with exogenous plasmid DNA molecules and thus not specific to pZα-Cas9 injection. It should be noted that plasmid pZα-Cas9 used in this study was designed for overexpression of muscle-specific Cas9 since Cas9 was driven by muscle-specific α-actin promoter. Therefore, we believe that injection of pZα-Cas9 might pose less risk of interfering with the development of other tissues. For example, the same phenotype of motor axon defect was found in all defective embryos (65% of 56 injected embryos) injected with pZα-Cas9 plus Pgk1 sgRNA. At the same time, however, no phenotype was found in more than 96% of 50 embryos injected with pZα-Cas9 plus Pgam2 sgRNA. This line of evidence suggested that the off-target effect caused by pZα-Cas9 hardly occurred in our study and, therefore, did not represent that the phenotypes we described here were due to off-target effect.

More recently, Yang et al. (2018) published a paper entitled “Generation of Cas9 transgenic zebrafish and their application in establishing an ERV-deficient animal model” (Biotechnol Lett, 40:1507–1518), in which they used Elongation Factor 1 (Elf1α) promoter to drive the overexpression of Cas9 around the whole body. They could inhibit target gene by injection sgRNA in F2 generation because they found that Cas9 overexpression alone did not affect the growth of zebrafish. I am suggesting that this evidence tends to support our use of Cas9 in this study.

Reviewer #2:

In the manuscript by Lin, et al., the authors identifiy Pgk1 as a novel secreted molecule that can stimulate neurite outgrowth. They further show that the secretion of Pgk1 is reduced when NogoA is expressed in the secreting cells, identify a potential mechanism by which NogoA reduces axon outgrowth independent of its interactions with Nogo receptors.While the identification of this novel role for Pgk1 is very interesting, there are a number of issues throughout the study that are of major concern, including a lack of important controls. In particular are the following points:1) Throughout the paper, there is inadequate information about "N"s in experiments, and inappropriate statistical analysis for multiple comparisons.

Thank you for pointing out these mistakes. In the revised manuscript, we present adequate percentages of experimental data among the total number of each “N” in all relevant sections and figure legends. Statistical analysis was also appropriately presented in each figure. Especially, we readjusted the representation of Western blot data, e.g., the relative quantification of phosphorylated proteins. We defined the expression level of control cell group, such as Sol8-vector CM or DM, as 1, and then we normalized it to the expression levels of experimental cell groups. Therefore, we deleted the p-value of the control group versus other experimental groups since there was no standard deviation (nonparametric) in the control group. Student *t*-test analysis was performed only for groups with standard deviation, such as the Sol8-NogoA CM group compared to the Sol8-NogoA CM + Pgk1 group. Statistical analyses were revised in Figures 1A, 1B, 1C, 2E, 5A, 5C, 5D and Figure 5—figure supplement 1D.

2) There is no indication of Pgk1 dose or timecourse throughout the entire study. Is there a dose dependent response?

We have performed a new experiment in which different doses of Pgk1 (0-, 11-, 33-, 66-, and 99-ng/ml) were added to NSC34 cells cultured in differentiation medium. As shown in the following Figure A, the effect of Pgk1 on the downregulation of p-Cofilin-S3 was dose-dependent (see new Figure 1—figure supplement 4). Moreover, we performed a time course experiment, in which NSC34 cells were treated with Pgk1 (33 ng/ml) for 0, 8, 16, 24 and 48 hrs. As shown in the following Figure B, the effect of Pgk1 on the downregulation of p-Cofilin-S3 was time-dependent from 8 through 24 hr (see new Figure 1—figure supplement 4). However, we noticed that p-Cofilin-S3 was unexpectedly increased if cells were treated with Pgk1 for 48 hr. The doses of Pgk1 we examined did not cause any negative effect on cell growth and survival. However, based on the above results and to ensure that we did not seriously affect cell physiology, we treated NSC34 cells with the minimal concentration (33-ng/ml of Pgk1 for 24 hr) throughout the entire study.

We added the above new data in the Results section as follows: “Next, to determine if Pgk1 alone is sufficient to induce NOM and neuronal differentiation, we switched to low trophic factor differentiation media (DM). […] Interestingly, the addition Pgk1 at that concentration not only rescued the number of neurite-bearing cells, but also enhanced neurite length of NSC34 cells cultured in Sol8-NogoA CM and DM (Figure 1D-E).”

Is the dose used physiologically relevant?

The Pgk1 dose (33 ng) used in this study was physiologically relevant and based on the concentration from what we learned in our preliminary investigation. Specifically, when we collected conditioned medium (CM) from culturing Sol8 cells transfected with Sol8-vector and Sol8-NogoA individually, we quantified Pgk1 contained in CM by the relative quantification method using commercially available standard Pgk1 protein. The result showed that the Pgk1 concentration was 31.80 ng/ml in Sol8-vector CM, while it was 14.97 ng/ml in Sol8-NogoA CM (see Author response image 1). Based on these data, we chose 33 ng/ml of Pgk1, which aligns with the result above for Sol8-vector CM, and decided to add it into DM for subsequent experiments in this study. Additionally, since the concentration of commercially available Pgk1 stock was 440 ng/ul, it was more convenient to make a 40X dilution to obtain final concentration of 11 ng/ul, followed by taking a volume of 3 ul to perform experiments. However, the actual effective concentration of Pgk1 in vivoneeds to be further analyzed.

3) Does the NogoA-Pgk1 secretion mechanism also function in primary muscle cells and motor neurons in culture? Is there evidence for NogoA over-expression inducing a reduction in Pgk1 expression in zebrafish and in mouse in vivo?

In response to your question, we performed a new experiment, in which we extracted blood samples from zebrafish and mouse and detected the expression level of Pgk1 in sera. Since it is very hard to extract blood from zebrafish and since its serum is very limited, we preferred to use WT and SOD1 G93A mice. The result demonstrated that the Pgk1 level in sera of SOD1 G93A mice at 90 days old was significantly lower than that found in the sera of WT mice at same age (see Figure 1—figure supplement 3). We also demonstrated that NogoA was overexpressed in the muscle of SOD1 G93A mice at 90 days old (see Figure 1—figure supplement 3A), which was consistent with the result reported by Bros-Facer et al. (2014). The above data deomonstrated in mouse in vivo system is strongly supportive for our finding that overexpression of NogoA in muscle cells results in a reduction in Pgk1 secretion.

As results shown above, we have demonstrated the NogoA-Pgk1 secretion mechanism functions in vivo, we, therefore, did not study further whether NogoA-Pgk1 secretion mechanism also function in primary muscle cells and motor neurons in culture.

We added the above results in the Results section and revised as follows: “We further employed 2D electrophoresis, followed by LC MS/MS, to analyze the total protein content in CM. […] Collectively, based on the above results provided in vitro and in vivo evidence, we chose Pgk1 for further study to confirm its potential role in NOM.”

4) At the end of the Results section, the authors mention anecdotally that there is a motor behavior phenotype and provide videos. However, there is no quantification and no description in the Materials and methods section to indicate any type of quantitative assessment of motor behavior. How many mice were used in these experiments?

We added more detail about motor behavior in the revised manuscript. In the Materials and methods section, we added the following text: “We also performed intramuscular injection of Pgk1 into the gastrocnemius muscle of the right hind leg of six 60-day-old (at P60) transgenic SOD1 G93A mice. […] The exercise capability value of each mouse was calculated from the increased fold(s) in the number of contractions between injected right leg versus untreated left leg. Data of each group were averaged from three mouse samples and presented as mean ± S.D.”

We also added some description in the Results section, as follows: “In the WT/PBS group, the muscle contraction of both hind legs was normal, exhibiting strong movement. […] Taken together, it is suggested that the Pgk1-injected right hind leg of SOD1 G93A mice could maintain some normal motor neurons to innervate muscle contraction.”

5) The authors conclude that Pgk1 functions independent of NgR. However, their only evidence for this is the lack of activation of downstream signaling cascades, and there are no positive controls included in these experiments. This is very indirect, and does not allow the authors to conclude Pgk1 function is independent of NgR.

In response to your question, we performed several new experiments to confirm whether the involvement of Pgk1 in pathway regulation was independent of NgR receptor activation by NogoA. Two study approaches were designed. First, we determined whether p-Cofilin expression was different between GST-Nogo66 addition and GST-Nogo66 plus Pgk1 addition to NSC34 cells cultured in differentiation medium (DM). Compared to the control group in which GST(100 ng/ml) was added in DM, NSC34 cells cultured in GST-Nogo66 (100 ng/ml) added to DM exhibited an increase of p-ROCK2 and p-EGFR expression and a decrease of p-Akt (please see Figure 5—figure supplement 3A), suggesting that the addition of GST-Nogo66 activates the NgR response pathway. Interestingly, the increased p-ROCK2 and p-EGFR and decreased p-Akt remained unchanged in NSC34 cells cultured in GST-Nogo66 plus Pgk1 addition (please see Figure 5—figure supplement 3A), suggesting that the addition Pgk1 did not affect the Nogo66/NgR pathway. However, compared to GST-Nogo66-treated cells, the GST-Nogo66 plus Pgk1-treated NSC34 cells exhibited reduced p-Cofilin expression through decreasing p-Limk-S323, but not through p-Limk-T508 (please see Figure 5—figure supplement 3B), suggesting that the reduced expression of p-Cofilin results from the presence of Pgk1 through some pathway other than Nogo66/NgR interaction.

Second, we determined whether p-Cofilin expression remained in a reduced state in NSC34 cells by the addition of Pgk1 into NSC34 cells cultured in DM when NgR receptor was blocked by NAP2 (NgR receptor antagonist peptide; 10 uM) (Sun et al., 2016). The addition of NAP2 in the culture caused a reduction of ROCK2 expression in NSC34 cells, a result consistent with that reported by Sun et al., 2016, indicating that NAP2 could block the NgR receptor (New Figure 5—figure supplement 3C). However, unlike increased ROCK2 in NSC34 cells cultured in NAP2 addition, ROCK2 expression remained unchanged in NSC34 cells cultured in NAP2 plus Pgk1 addition (New Figure 5—figure supplement 3D). In contrast, the expression of p-Pak1/p-P38/p-MK2/p-Limk1-S323/p-Cofiln axis was reduced (New Figure 5—figure supplement 3D).

Taken together, we suggest that the reduction of p-Cofilin mediated by ePgk1 does not occur through Nogo66/NgR interaction in neuronal cells since ePgk1 addition can still functionally reduce the p-Pak1/p-P38/p-MK2/p-Limk1-S323/ p-Cofiln axis in the absence of NgR receptor of neuronal cells.

Based on above new data, it should strengthen our conclusion that ePgk1 reduces the expression of p-Cpfilin in NSC34 cells to enhance neurite outgrowth of motor neurons is independent of Nogo66/NgR pathway. We added these new data in the Results section under the second paragraph of subtitle entitled The signaling pathway underlying the involvement of ePgk1-mediated reduction of p-Cofilin-S3 as follows:

“To confirm whether Pgk1 involved regulating pathway is independent of activation of NgR receptor by NogoA. […]And, the molecular pathway ePgk1 involved in NOM is independent of the Nogo66/NgR interaction.”

6) How NogoA reduces the secretion of Pgk1 from muscle cells and how extracellular Pgk1 inhibits the intracellular Rac1-PAK pathway is unclear and needs additional investigation and/or discussion.

In response to your question, we added some statements in the Discussion as follows: “To address the reduced secretion of Pgk1 from muscle cells by NogoA, as well as inhibition of the neuronal intracellular Rac1 pathway by ePgk1, we turn to the literature. […] The reduced ePgk1 may then cause a decrease in extracellular proteins, such as angiostatin, which, in turn, would increase the activity of Rac1, finally causing a physiological microenvironment for motor neurons that would not favor neurite outgrowth.”

Additional comments on the specific Figures:Figure 1:The authors need to confirm that there is no NOGO-A in the conditioned media

In response of this question, we confirmed the absence of NogoA in the condition medium cultured by Sol8-NogoA cells. We added these data in the Results section as follows: “Importantly, NogoA was not detected in the CM cultured by Sol8-NogoA cells after Dox induction for 48 hr (Figure 1—figure supplement 1C). This line of evidence suggested that the component(s) contained in CM from cultured Sol8-NogoA play(s) a role in NOM inhibition, but not through NogoA contained in medium”.

The specificity of the Pgk1 Ab used in Figure 1B needs to be confirmed

We have increased the amount of total proteins from 4 to 20 ug for Western blot analysis. Similar to the previous 4 ug data the result demonstrated that only one sharp positive band was exclusively shown on the gel when Pgk1 antibody (Abcam: ab38007) was used, suggesting that the Pgk1 antibody used in this study presents sufficiently high specificity to detect Pgk1. (Please see newly added Figure 1—figure supplement 2B shown).

Figure 2:It is unclear how well the markers used (MAP2, ChAT, GAP43, SYN1) can actually reflect a differentiated status in NSC34 cells. These cells have very small processes, and therefore are unlikely to represent a truly differentiated state.

We followed a paper published in *Neurochemistry International* (2013, 62: 1029-38) by Maier et al., entitled “Differentiated NSC34 motoneuron-like cells as experimental model for cholinergic neurodegeneration.” The authors employed markers of MAP2, ChAT, GAP43, and SYN1 to define differentiation of NSC34 cells and stated that the increased expression of these proteins served as indicators of NSC34 differentiation. Although the expression of these proteins is unlikely to represent a truly differentiated state, as you pointed out, we used the intracellular markers MAP2, ChAT, and GAP43, as well as Syn1-labeling signal in the growth cones’ terminus, as a guidepost in determining whether NSC34 cells further develop toward differentiation and maturation or whether these cells maintain a proliferative state when cultured in Sol8-vector CM, Sol8-NogoA CM and Sol8-NogoA CM plus Pgk1.

In subsection “Supplementary addition of Pgk1 can enhance the maturation of NSC34 cells cultured in Sol8-NogoA CM”, the authors state "The number of neurons with Syn1 signal at the end of synapse could be restored to that cultured in Sol8-vector…". It is unlikely that these are synapses, they look more like growth cones and don't contact "postsynaptic" structures

Thank you for pointing out this mistake. We corrected it as follows: “….The number of neurons with Syn1 signal at the end of growth cones could be restored to that cultured in Sol8-vector…”

Figure 4:In zebrafish experiments, the authors use CRISPR/Cas9 system to knockout Pgk1 and Pgam2 from muscle cells. There needs to be data validating that sgRNAs targeting Pgk1 and Pgam2 are effective in reducing Pgk1 and Pgam2 expression, respectively. Similarly, in the muscle-specific over expression paradigm, the authors need to show that Pgk1 expression is increased.

In response to your suggestion, we added Crispr/cas9-mediated inactivation and upregulation of Pgk1 in zebrafish muscle cells in the Results section, as follows: “We used muscle-specific α-actin promoter to drive overexpression of Cas9 in muscle cells in order to exclusively knock out the Pgk1 gene. […] As suggested by quantification of the axonal extension phenotype shown in Figure 4—figure supplement 2, the increase of Pgk1 in muscle cells enhances NOM.”

While the authors provide information on the penetrance of the phenotypes, they do not quantify the extent of axonal extension in any meaningful way.

Thank you for your suggestion. In the revised manuscript, we quantified the percentage of axonal extension phenotype, presented a new Figure 4—figure supplement 2, and added these data in the Results section as follows: “When these RFP-expressing cells were sorted and examined by Western blot analysis, the results demonstrated that Pgk1 was overexpressed (Figure 4—figure supplement 1). Meanwhile, motoneuron axons exhibited ectopic growth (Figure 4G). As suggested by quantification of the axonal extension phenotype shown in Figure 4—figure supplement 2, the increase of Pgk1 in muscle cells enhances NOM.”

Figure 5:The authors conclude that the effect of Pgk1 is NgR-independent due to the lack of p-ROCK2-Y256, p-EGFR-Y1173 and p-Akt-S473 (Figure 5—figure supplement 1). However, there is no positive control in this experiment. The authors should show that they are able to detect changes in these pathways in response to NgR activation by NOGO-A.

In response your question, we designed a positive control, in which Nogo66 fused with GST (GST-Nogo66) was used as a positive control, to determine whether p-Cofilin expression was any different between GST-Nogo66 addition and GST-Nogo66 plus Pgk1 addition to NSC34 cells cultured in DM. The results were shown in new Figure 5 and Figure 5—figure supplement 3A-B.

Compared to the control group in which GST(100 ng/ml) was added to DM, NSC34 cells cultured in GST-Nogo66 (100 ng/ml) added to DM exhibited increased p-ROCK2-Y256 and p-EGFR-Y1173 expression and decreased p-Akt-S473 expression (please see figures related to #5 above, as well as new Figure 5—figure supplement 3A), suggesting that GST-Nogo66 addition activated the NgR response pathway (served as positive control). Interestingly, the increased p-ROCK2-Y256/p-EGFR-Y1173 and decreased p-Akt-S473 remained unchanged in NSC34 cells cultured in GST-Nogo66 with Pgk1 addition (see Figure 5—figure supplement 3A), suggesting that the addition of Pgk1 did not affect the Nogo66/NgR pathway. However, compared to GST-Nogo66-treated NSC34 cells, the GST-Nogo66 plus Pgk1-treated NSC34 cells exhibited a reduction of p-Cofilin-S3 expression through a decrease in p-Limk-S323, but not through p-Limk-T508 (see figures shown for #5 and new Figure 5—figure supplement 3B), suggesting that the reduced expression of p-Cofilin results from the presence of Pgk1 through a pathway other than Nogo66/NgR interaction. Based on this line of evidence, we conclude that the effect of ePgk1 on the reduction of p-Cofilin in neuronal cells is NgR-independent.

Figure 6:Because the authors did not provide information on the dose-dependency and timecourse of Pgk1 in their in vitro experiments, it is unclear why they chose the specific dose and timing regimens for their in vivo experiments.

In response to your question, we performed a new experiment in which different doses of Pgk1 (0-, 11-, 33-, 66-, and 99-ng/ml) were added to NSC34 cells cultured in differentiation medium. As shown in the following Figure A, the effect of Pgk1 on the downregulation of p-Cofilin-S3 was dose-dependent (see new Figure 1—figure supplement 4). Moreover, we performed a time course experiment, in which NSC34 cells were treated with Pgk1 (33 ng/ml) for 0, 8, 16, 24 and 48 hrs. As shown in the following Figure B, the effect of Pgk1 on the downregulation of p-Cofilin-S3 was time-dependent from 8 through 24 hr (see new Figure 1—figure supplement 4). However, we noticed that p-Cofilin-S3 was unexpectedly increased if cells were treated with Pgk1 for 48 hr. The doses of Pgk1 we examined did not cause any negative effect on cell growth and survival. However, based on the above results and to ensure that we did not seriously affect cell physiology, we treated NSC34 cells with the minimal concentration (33-ng/ml of Pgk1 for 24 hr) throughout the entire study.

We added the above new data in the Results section as follows: “Next, to determine if Pgk1 alone is sufficient to induce NOM and neuronal differentiation, we switched to low trophic factor differentiation media (DM). […] Interestingly, the addition Pgk1 at that concentration not only rescued the number of neurite-bearing cells, but also enhanced neurite length of NSC34 cells cultured in Sol8-NogoA CM and DM (Figure 1D-E).”

Fish were injected with 2ug Pgk1, then placed in Dox+ water for 1 week. How long is the injected Pgk1 biologically active?

In response to your question, we performed a new experiment, in which we injected 2 ug of GFP (control group) or Pgk1 (experimental group) into transgenic zebrafish line Tg(*TetON-Rtn4al*), followed by immersing zebrafish in water containing Dox, to induce overexpression of Rtn4al/NogoA in muscle. In the control group, as immersion time increased from one, two and three weeks, the degree of NMJ denervation also increased. However, compared with the GFP-injected control group, delay in NMJ denervation was still observed in the Pgk1-injected group at two weeks after injection. Nevertheless, no significant difference was observed in the extent of NMJ denervation between control and experimental groups at three weeks after injection. This line of evidence suggested that adult zebrafish muscle injected Pgk1 (2 ug) was able to retain its biological activity for two weeks post-injection. We added this information in the Results and attached a new figure in the revised version.

We added these data in the Results section as follows: “While we observed denervation at NMJ after Rtn4al/NogoA induction in adult zebrafish (Figure 6—figure supplement 1), as evidenced by reduced colocalization of presynaptic and postsynaptic markers, this defect saw much improvement by addition of Pgk1, but not GFP control, through intramuscular injection (Figure 6A-G), suggesting ePgk1 supports NMJ integrity. Although adult zebrafish muscle injected with Pgk1 did exhibit delayed NMJ denervation, it could still retain its normal biological activity for two weeks after a single shot (Figure 6—figure supplement 2).”

There is a discrepancy between what is in the Results section and the Materials and methods section with respect to the experiments done in mice in vivo. The authors indicate that mice were injected with Pkg1 once every 15 days between P60 and P120. However, in the Materials and methods section (546-548) they indicate tissue sections were taken at P75. Is the data presented in Figure 6 H-N from P75 or P120 mice?

We apologize for this. The mice shown in Figures 6 H-N were injected with Pgk1 on the right hind leg at P60, followed by performing frozen section and immunofluorescence staining to examine the innervated NMJ at P75. This was what we described in the Materials and methods section (546-548). While mice shown in the video were injected with Pgk1 on the right hind leg at P60, another Pgk1 injection was continuously administered every 15 days up to P120. When they were at P130, we recorded their movement by video (Figure 6—video 1-6) and evaluated their muscle contraction capability (Figure 6—figure supplement 5).

To make this description clear, we added more detail in the Materials and methods section as follows: “We also performed intramuscular injection of Pgk1 into the gastrocnemius muscle of the right hind leg of six 60-day-old (at P60) transgenic SOD1 G93A mice. […] Data of each group were averaged from three mouse samples and presented as mean ± S.D. “

We also added some description in the Results section, as follows: “In the WT/PBS group, the muscle contraction of both hind legs was normal, exhibiting strong movement. […] Taken together, it is suggested that the Pgk1-injected right hind leg of SOD1 G93A mice could maintain some normal motor neurons to innervate muscle contraction.”

The authors should provide images and other metrics (neurite outgrowth, expression of motor neuron specific genes) from their iPS-SOD G85R motor neuron cultures so the degree of differentiation these iPS cells can be assessed.

In response to your suggestion, we provided more detailed information and added new images in new *Figure 6—figure supplement 3*. As shown in this figure, after 14 days of motor neuron induction, the iPSC-SOD1G85R cells expressed the pluripotency-specific markers Oct4, Nanog and SSEA4 and then began to differentiate motor neurons. After 15 days of motor neuron differentiation, more than 90% of induced cells expressed the neural stem cell-specific markers sox1 and N-cadherin, as well as the motor neuron precursor-specific markers Oligo2 and Islet1. After 27 days of motor neuron differentiation, more than 90% of induced cells expressed motor neuron-specific protein HB9 and nerve fiber protein neurofilament. Therefore, our results demonstrated the highly efficient differentiation of motor neurons from iPSCs harboring a G256C point mutation (G85R on peptide sequence) on human SOD1 gene in an ALS patient.

- Figure 6—figure supplement 2 panel B: Why is there no change in the levels of p-Cofilin in response to Pgk1 in the control cells? In Figure 1A, this same treatment causes a ~3-fold decrease p-Cofilin levels.

We apologize for this mistake. We repeated this experiment and replaced this figure by a new one. (see new Figure 6—figure supplement 4B)

Reviewer #3:

[…] This study revealed a new indirect mechanism how NogoA in the muscle affected neurite growth of motoneurons. The finding is interesting and helpful for understanding the probable mechanisms in steering axon growth. However, there are some issues needed to be addressed.1) Nogo-A is a transmembrane protein, it could exert its effects by direct interacting with its receptors on the neurons, or it can work indirectly through affecting other molecule expression or secretion in the muscle. The authors used CM from NogoA-overexpressing cells to treat neurons, which supports the indirect effect. Given NogoA is a transmembrane protein, it is not likely present in the CM. Without the co-culture experiment of muscle cells and motor neurons, it's inappropriate to exclude the possibility that NogoA in the muscle could also affects neurite growth through a direct ligand/receptor interaction. Therefore, the conclusion "In this study, we revealed that overexpression of NogoA in muscle inhibits NOM through decreasing Pgk1 secretion (ePgk1), but without inducing the canonical Nogo66/NgR pathway."(subsection “Intramuscular injection of Pgk1 is able to rescue NMJ denervation caused by Rtn4al-overexpression transgenic zebrafish and SOD1 G93A transgenic mice”) is not accurate.

We believe our conclusion is accurate. In this study, NSC34 cells were treated with CM from cultured NogoA-overexpressing muscle cells, but no NSC34 cells were directly cultured with NogoA-overexpressing muscle cells. Furthermore, we could not detect NogoA protein in CM (see Figure 1—figure supplement 1C). Thus, it is reasonable to rule out the possibility that NogoA on the muscle cell membrane interacts directly with NgR receptor on the neuronal cells through ligand/receptor interaction to enhance neurite growth, as you speculated.

Moreover, we designed two new study strategies to support our hypothesis. First, we determined whether p-Cofilin expression was different between GST-Nogo66 addition and GST-Nogo66 plus Pgk1 addition to NSC34 cells cultured in differentiation medium (DM). Compared to the control group in which GST (100 ng/ml) was added in DM, NSC34 cells cultured in GST-Nogo66 (100 ng/ml) added to DM exhibited an increase of p-ROCK2 and p-EGFR expression and a decrease of p-Akt (please see Figure 5—figure supplement 3A), suggesting that the addition of GST-Nogo66 activates the NgR response pathway (served as positive control). Interestingly, the increased p-ROCK2 and p-EGFR and decreased p-Akt remained unchanged in NSC34 cells cultured in GST-Nogo66 plus Pgk1 addition (please see Figure 5—figure supplement 3A), suggesting that the addition of Pgk1 did not affect the Nogo66/NgR pathway. However, compared to GST-Nogo66-treated cells, the GST-Nogo66 plus Pgk1-treated NSC34 cells exhibited reduced p-Cofilin expression through decreasing p-Limk-S323, but not through p-Limk-T508 (please see Figure 5—figure supplemental 3B), suggesting that the reduced expression of p-Cofilin results from the presence of Pgk1 through some pathway other than Nogo66/NgR interaction.

Second, we determined whether p-Cofilin expression remained in a reduced state in NSC34 cells by the addition of Pgk1 to NSC34 cells cultured in DM when NgR receptor was blocked by NAP2 (NgR receptor antagonist peptide; 10 uM) (Sun et al., 2016). The addition of NAP2 in the culture caused the reduction of ROCK2 expression in NSC34 cells, a result consistent with that reported by Sun et al., 2016, indicating that NAP2 could block the NgR receptor. However, unlike increased ROCK2 in NSC34 cells cultured in NAP2 addition, ROCK2 expression remained unchanged in NSC34 cells cultured in NAP2 plus Pgk1 addition (New Figure 5—figure supplement 3C). In contrast, the expression of p-Pak1/p-P38/p-MK2/p-Limk1-S323/p-Cofilin axis was reduced (New Figure 5—figure supplement 3D).

Taken together, we suggest that the reduction of p-Cofilin mediated by ePgk1 does not occur through Nogo66/NgR interaction in neuronal cells since ePgk1 addition can still functionally reduce the p-Pak1/p-P38/p-MK2/p-Limk1-S323/p-Cofilin axis in the absence of NgR receptor of neuronal cells. In sum, ePgk1 triggers a reduction in p-Cofilin-S3, in turn promoting NOM through decreasing a novel p-Pak1-T423/p-P38-T180/p-MK2-T334/p-Limk1-S323 axis *via* reducing Rac1-GTP activity in neuronal cells. Therefore, we conclude that 1) a cross-tissue mediator, i.e., Pgk1 secreted from muscle cells, was found; (2) this extracellular Pgk1 enhances neurite growth of motor neurons through a novel signal pathway which is distinct from the well-known NogoA(Nogo66)/NgR pathway between muscle and neuronal cells or among neuronal cells; (3) overexpression of NogoA in muscle cells causes the decrease of secreted Pgk1, resulting in the increase of p-Cofilin in neuronal cells, which, in turn, inhibits neurite outgrowth of motor neurons, suggesting that neurite outgrowth of motor neurons can also be negatively regulated by reducing the amount of secreted Pgk1 from NogoA-overexpressing muscle cells; and, finally, (4) the effect of ePgk1 on neuronal cells is NgR-independent.

2) Is there any evidence in vivo that NogoA upregulation in the muscle affected its Pgk1 production, i.e. the NogoA and Pgk1 levels in the SOD1 mutant mice or Rtn4al-overexpressing transgenic zebrafish? Or at least confirm whether overexpressing NogoA in the muscle of normal adult mice using viral vector will also affect Pgk1 secretion.

In response to your question, we performed a new experiment, in which blood samples from 90-day-old WT and SOD1 G93A mice were extracted to detect the expression level of Pgk1 in sera. The Pgk1 level in SOD1 G93A mice sera was significantly lower than that in the sera of WT mice (see Figure 1—figure supplement 3). In addition, NogoA was overexpressed in the muscle of SOD1 G93A mice, which was consistent with the result reported by Bros-Facer et al., 2014. These results provide in vivo evidence that NogoA upregulation in muscle cells negatively affects the secretion of Pgk1.

We added the above results in the Results section and revised as follows: “We further employed 2D electrophoresis, followed by LC MS/MS, to analyze the total protein content in CM. Among these examined proteins, we found that the level of Pgk1 protein contained in Sol8-NogoA CM was significantly reduced compared to that in Sol8-vector CM (Figure 1—figure supplement 2). […] Collectively, the above results provided both in vitro and in vivo evidence of the reduction of Pgk1 secretion in NogoA-overexpressed muscle; therefore, we chose Pgk1 for further study to confirm its potential role in NOM.”

3) The authors seemed to confuse with neurite and synapse: subsection “in vitro platform of neurite outgrowth of motor neurons derived from NSC34 cells”; "Synapses extending from each cell were captured by microscope.…"; "calculate the length of synapses…"

Thank you for pointing out this mistake. We made the following change: “Neurites extending from each cell were captured by microscopy (Leica) under 200 times magnification and labeled by Neurolucida Neuron Tracing Software 9.0 (Neurolucida 9.0). […] To count the number of neurite-bearing cells, we calculated the number of cells with neurites at least 50 μm in length in each range.”

4) In Figure 1—figure supplement 2, the authors need quantify the Pgk1 amount in the CM. In addition, there is no information about the source and the concentration of ePgk1 added into the culture. Whether is it comparable to the amount in the CM?

In response to your question we quantified the amount of Pgk1 in CM. When we collected conditioned medium (CM) from culturing Sol8 cells transfected with Sol8-vector and Sol8-NogoA individually, we quantified Pgk1 contained in CM by the relative quantification method using commercially available standard Pgk1 protein. Results showed that the Pgk1 concentration was 31.80 ng/ml in Sol8-vector CM, while it was 14.97 ng/ml in Sol8-NogoA CM. Based on these data, we chose 33 ng/ml of Pgk1, a choice which aligns well with the result above for Sol8-vector CM, and decided to add it into DM for subsequent experiments in this study. Additionally, since the concentration of commercially available Pgk1 stock was 440 ng/ul, it was more convenient to make a 40X dilution to obtain a final concentration of 11 ng/ul, followed by taking a volume of 3 ul to perform experiments.

5) In the method of neurite outgrowth (subsection “in vitro platform of neurite outgrowth of motor neurons derived from NSC34 cells”), it says: "In each image, 40~50 400 cells were randomly chosen to calculate the length of synapses and the number of neurite-bearing cells, the synapses of which were at least 50 μm in length.". If only cells with neurite length longer than 50 μm were chosen, why the average neurite lengths in some groups of Figure 1E are shorter than 50 μm?

We apologize for the confusion. To make this description clear, we profiled the neurite length distribution of each group. Specifically, after we measured the length of neurites of all 120 examined cells in each group, we grouped their size into several groups, such as every 25 intervals under 50 um, every 50 intervals under 200 um, 201-300 and 301-600 um. To know the number of neurite-bearing cells, we marked a line on the figures to indicate the neurite length which was more than 50 μm (see Figures 1D-E).

In the revised manuscript, we added more detailed information to the Materials and methods section, as follows: “Neurites extending from each cell were captured by microscopy (Leica) under 200 times magnification and labeled by Neurolucida Neuron Tracing Software 9.0 (Neurolucida 9.0). For each image, 40 cells cultured in either CM or DM were randomly chosen to measure the length of neurites. Data were obtained from three independent trials (120 cells in total). To profile neurite length distribution, we grouped neurite length into several groups, namely every 25 intervals under 50 um, every 50 intervals under 200 um, and 201-300 and 301-600 um, and calculated the number of cells in each range. To count the number of neurite-bearing cells, we calculated the number of cells with neurites at least 50 μm in length in each range.”

In the Results section, we added the following: “Interestingly, the addition Pgk1 at that concentration not only rescued the number of neurite-bearing cells, but also enhanced neurite length of NSC34 cells cultured in Sol8-NogoA CM and DM (Figure 1D-E).”